# Decoupling tRNA promoter and processing activities enables specific Pol-II Cas9 guide RNA expression

David J.H.F. Knapp [1], Yale S. Michaels [1], Max Jamilly[1], Quentin R.V. Ferry[1], Hector Barbosa[1], Thomas A. Milne [2] & Tudor A. Fulga[1]

Spatial/temporal control of Cas9 guide RNA expression could considerably expand the utility of CRISPR-based technologies. Current approaches based on tRNA processing offer a promising strategy but suffer from high background. Here, to address this limitation, we present a screening platform which allows simultaneous measurements of the promoter strength, 5′, and 3′ processing efficiencies across a library of tRNA variants. This analysis reveals that the sequence determinants underlying these activities, while overlapping, are dissociable. Rational design based on the ensuing principles allowed us to engineer an improved tRNA scaffold that enables highly specific guide RNA production from a Pol-II promoter. When benchmarked against other reported systems this tRNA scaffold is superior to most alternatives, and is equivalent in function to an optimized version of the Csy4-based guide RNA release system. The results and methods described in this manuscript enable avenues of research both in genome engineering and basic tRNA biology.

[1] Weatherall Institute of Molecular Medicine, Radcliffe Department of Medicine, University of Oxford, Oxford OX3 9DS, UK. [2] Weatherall Institute of Molecular Medicine, MRC Molecular Haematology Unit, NIHR Oxford Biomedical Research Centre Programme, University of Oxford, Oxford OX3 9DS, UK. Correspondence and requests for materials should be addressed to D.J.H.F.K. (email: david.knapp@ndcls.ox.ac.uk) or to T.A.F. (email: tudor.fulga@imm.ox.ac.uk)

Most CRISPR/Cas9 guide RNA (gRNA) expression systems use RNA polymerase-III (Pol-III) promoters such as U6 [1,2]. While highly efficient, these promoters act in a constitutive fashion[3]. Although constitutive gRNA expression is compatible with many genome editing applications, independent control of editing events in multiple tissues or at different times requires inducible gRNA expression. However, most expression systems enabling spatial/temporal control of promoter activity rely on Pol-II-mediated transcription. Following transcription, Pol-II products are rapidly modified with a 5′ cap and poly-A tail and exported from the nucleus. These modifications and altered localization could prevent efficient use of Cas9 gRNAs[4]. Consequently, a number of strategies have been proposed to excise gRNAs from Pol-II transcripts. These include the use of alternative transcriptional terminators[5], embedding the gRNA in a spliced intron[6], self-cleaving ribozymes-based release systems[4,7], and the use of Csy4 or orthologous ribonucleases[4]. These strategies, however, suffer from relatively poor activation rates downstream of Pol-II promoters, or require the addition of toxic[4] and potentially immunogenic proteins. Thus, there remains a need for the development of effective nonconstitutive CRISPR/Cas9 gRNA expression systems.

tRNAs represent a highly conserved class of RNA molecules that are recognized and precisely processed by RNase P and RNase Z[8]. Various tRNAs have been exploited to allow polycistronic gRNA production with high processing efficiencies[9,10]. tRNAs, however, contain internal Pol-III promoters[8]. Indeed, tRNAs have been used to replace U6 promoters for gRNA production[11], albeit at somewhat lower efficiency[12]. Intriguingly, previous studies reported Pol-II-specific gRNA activity using tRNA-based multiplexing systems[7,13]. However, in one instance the Cas9 was also placed under inducible control[13] and thus the gRNA may still have been constitutively expressed. In addition, this work was carried out in *Drosophila* that the authors suggest has low intrinsic tRNA promoter strength, and thus may not translate to mammalian cells[13]. Even in flies, the requirement of having both Cas9 and the gRNA under the same inducible promoter precludes the use of this system for multiple editing events with different timing of initiation. The second study employed two gRNAs flanked by ribozymes, and the detection system relied on releasing both gRNAs[7]. In this case, the first gRNA was upstream of the tRNA and thus not constitutively expressed by the tRNA Pol-III promoter activity. While such a system could potentially allow Pol-II specificity in some cases, this approach would be difficult to generalize.

The regions involved in tRNA promoter and processing activity have been previously identified[14–16]. While most positions overlap, we hypothesized that the differential requirements for these processes (DNA sequence identity and RNA structure for promoter and processing, respectively) might enable their decoupling, and thus provide an opportunity to re-engineer a tRNA scaffold with optimal parameters for gRNA release. To test this hypothesis, we performed a mutational screen on the human tRNA$^{Pro}$ (AGG; tRNAscan-SE ID: chr1.trna58) and independently measured the effects of base substitutions on promoter, as well as 5′ and 3′ processing activities. Based on this screen, we engineered tRNA variants that have no detectable promoter activity but retain sufficient processing to allow specific Pol-II-dependent Cas9 gRNA production, and demonstrate their superiority to most alternative Pol-II release systems.

## Results

**tRNAs have strong endogenous Pol-III activity in human cells.** First, we investigated the transcriptional activity, 3′ processing ability, and functional gRNA production in human cells of several wild-type tRNAs which have been previously used for gRNA multiplexing or Pol-II expression[7,9,11,13] (Fig. 1). Quantitative PCR (qPCR) of gRNAs placed downstream of native tRNAs revealed robust constitutive gRNA production in the absence of external promoters in all cases, albeit at lower overall levels than U6-driven gRNA expression (≥92% probability two-sided, Bayesian Estimation Supersedes the *t* test (BEST) test[17,18], Fig. 1c). This is consistent with findings that tRNA promoters appear to be slightly less efficient than U6 for gRNA production[12]. All human and fly tRNAs tested showed very efficient 3′ processing activity as measured using a modified circularization assay[9] (Fig. 1d, Supplementary Fig. 1). We next performed functional assays using a reporter system that expresses enhanced cyan fluorescence protein (ECFP) in the presence of a cognate gRNA[4,19], mitigating many biological (e.g. epigenetic state, microRNA) and technical (e.g. nonspecific antibody binding) confounding factors. This provides a sensitive snapshot measurement of the amount of functional gRNA produced at the single cell level in a defined time window. Importantly, with the exception of rice tRNA$^{Gly}$, all tRNAs tested in this assay enabled efficient transcriptional activation at levels equivalent to U6, in the absence of any additional Pol-II or Pol-III promoters (Fig. 1e, f, Supplementary Fig. 2a). Notably, fly and rice tRNA$^{Gly}$ showed promoter activity, processing ability, and functional gRNA production in human cells, albeit the values displayed by rice tRNA$^{Gly}$ were reduced compared to human tRNA$^{Gly}$ (two-sided, BEST probabilities of decreased effect 89%, Fig. 1, Supplementary Fig. 1 and 2a). These results reflect the strong conservation of tRNA systems across kingdoms and suggest that the principles described in this study will likely be applicable to a number of model organisms. Critically, these results confirm that tRNAs alone produce functional gRNAs constitutively and independent of external promoters, making them unsuitable for generalizable spatial/temporal controlled expression.

**Base dependencies of tRNA promoter and processing activities.** To test whether the processing and promoter activities of human tRNAs could be dissociated, we designed a variant screening strategy using the human tRNA$^{Pro}$ backbone. This entailed generation of high-content libraries in which each construct represented a single variant tRNA$^{Pro}$ flanked by a pair of gRNAs (Fig. 2a, Supplementary Fig. 3). The regions subjected to mutations were chosen based on their involvement in promoter and processing activities, as well as their lack of secondary structure determinants[14–16]. Two parallel libraries were generated, of which one contained an upstream Pol-II CMV promoter and one did not ((+)CMV or (−)CMV). Using an RNA circularization-nested RT-PCR protocol (Fig. 1, Supplementary Fig. 1, Online Methods), we then sequenced both the pDNA library and the circular RNA (circRNA) products (Fig. 2a). Quantitative analysis of barcode reads in the processed and unprocessed fractions provided an estimate of processing activity, while comparing the abundance of each mutation in the circRNA and plasmid pool allowed an estimate of promoter strength (Fig. 2a).

Analysis of promoter strength revealed that most promoter inactivating mutations resided in the D-loop, although position- and even nucleotide-specific effects were observed across all variable sites (Fig. 2b, c). In contrast, only a few specific mutations in the T-loop were detrimental to promoter activity, while others seemed to increase it (Fig. 2b, c). With regard to 3′ processing, most (but not all) mutations in the T-loop had strong detrimental effects on processing, while mutations in the D-loop appeared to have a lesser impact (Fig. 2d, e). Estimations of 5′ processing were hampered by barcode degradation in these libraries, presumably due to the decapping reaction. Therefore,

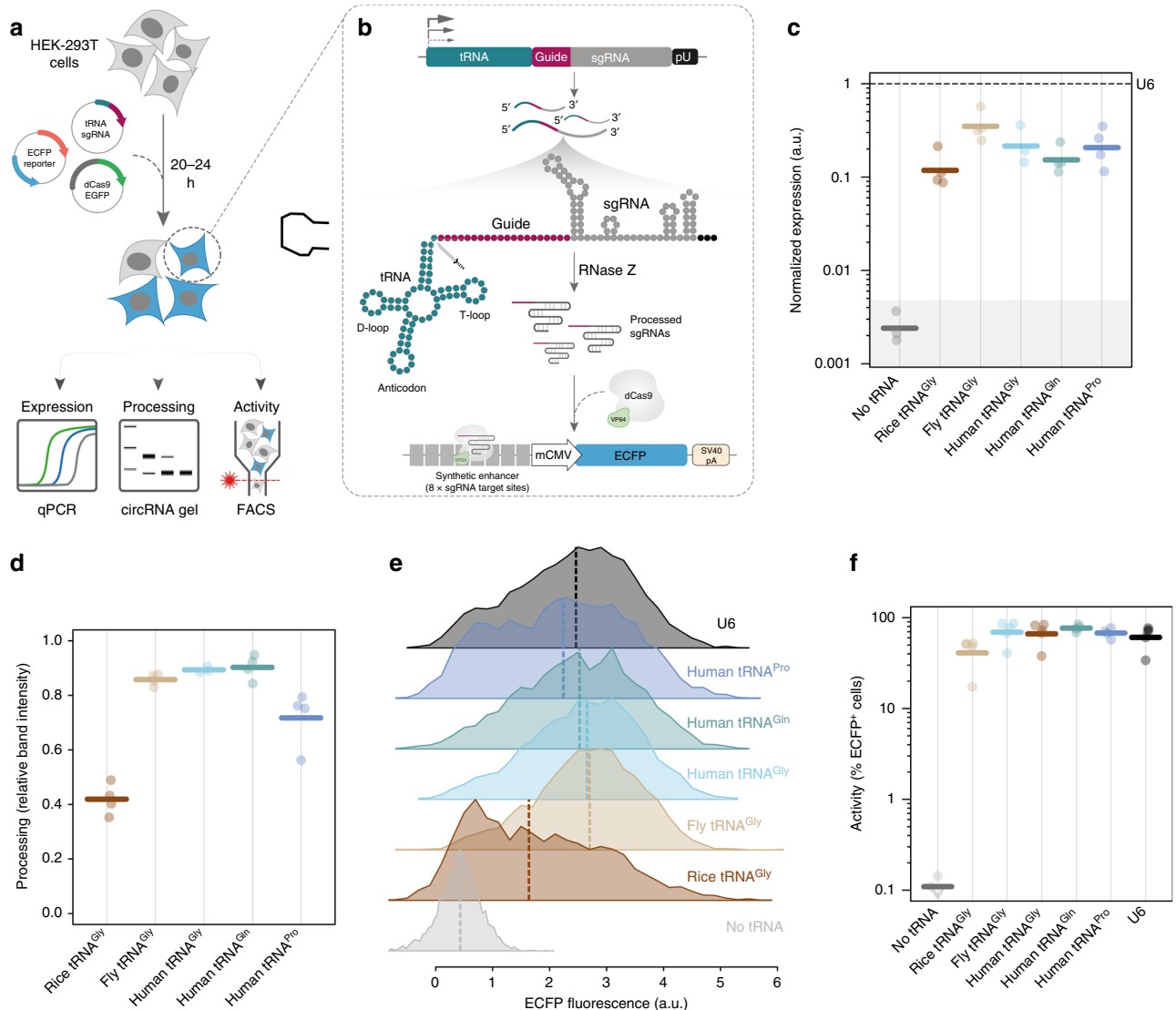

**Fig. 1** Wild-type tRNAs display strong Pol-III promoter activity and gRNA production. **a** Experimental strategy for testing functional gRNA production downstream of tRNAs. **b** Schematic diagram of molecular events occurring in cells transfected with tRNA-sgRNA constructs. **c** gRNA expression as measured by qPCR relative to Cas9 and U6 ($\Delta\Delta$Ct) ($n = 4$ independent experiments, $n = 3$ for no promoter control; dashed line = gRNA levels for U6). Shaded area represents the 75% credible mass (Bayesian Estimation Supersedes the $t$ test (BEST) test[17, 18]) for the no tRNA control. **d** 3′ processing ability of each wild-type tRNA tested. Efficiency represents the ratio of band intensity between the unprocessed and processed bands on a 2% agarose gel following RNA circularization and nested RT-PCR (thick lines = mean values). **e** Representative flow cytometry histograms of reporter levels downstream of U6 and various tRNA promoters. Values represent asinh(ECFP$_{fluoresence}$/150) (thick lines = median fluorescent intensities). **f** Percent reporter ECFP + cells within transfected cells (thick lines = geometric mean values; $n = 5$ independent experiments; hollow downward triangles = points at or below the limit of detection). Source data are provided as a Source Data file

only a partial barcode could be recovered, limiting the number of reads available for analysis and artificially lowering processing estimates. Nevertheless, values were obtained for most single nucleotide variants. This identified several specific nucleotides both in the D- and T-loops which appear to affect 5′ processing (Fig. 2f, g). Interestingly, there was little correlation between mutations affecting 3′ processing, 5′ processing, and promoter activity, supporting our hypothesis that these activities could be dissociated to some degree (Fig. 2h).

**Engineering tRNA scaffolds for Pol-II gRNA expression.** Based on the results of our mutagenesis screen, we selected pairs of mutations which should maximally decrease promoter strength while minimally affecting processing ability, as well as pairs

which should inhibit processing but not affect promoter strength (Fig. 3a). Since most promoter-detrimental mutations mapped to the D-loop and these tended to have lesser effects on processing, we also created a minimal tRNA backbone by completely deleting the D-loop and the anticodon ($\Delta$tRNA$^{Pro}$, Fig. 3b). This architecture is supported by previous reports suggesting that a similar minimal scaffold retains processing activities equivalent to wild-type tRNAs in *Drosophila*[15]. Analysis of the 3′ processing efficiency revealed that all selected double mutants lost their activity to some degree (Supplementary Fig. 4a, b). Surprisingly, in the $\Delta$tRNA$^{Pro}$ scaffolds, which retained enough promoter activity to be detectable, 3′ processing was not decreased compared to wild type (Supplementary Fig. 4a, b).

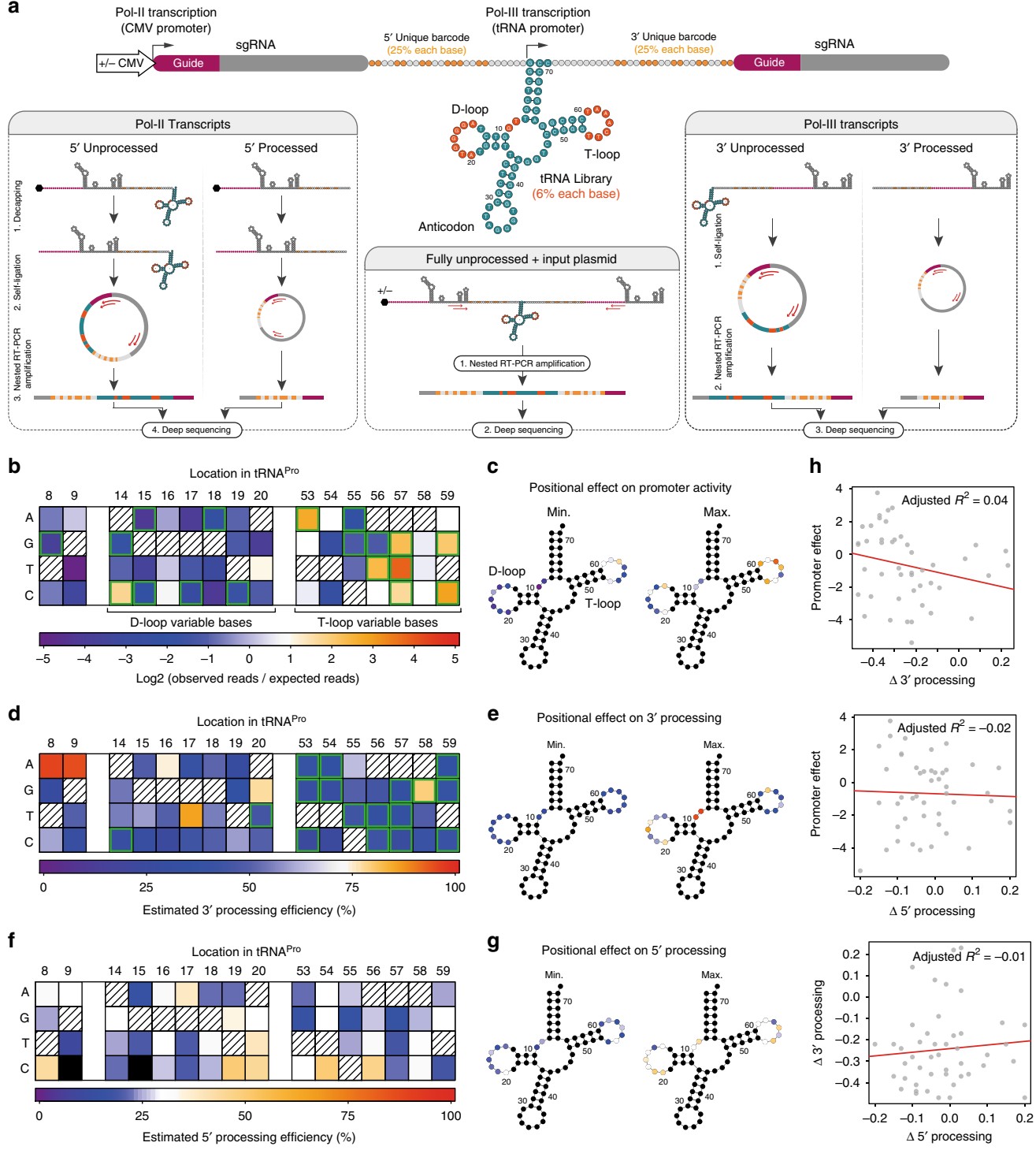

Measurement of promoter activity by qPCR revealed a slight effect of the double mutants designed to decrease promoter strength compared to wild type (green tones in Supplementary Fig. 4c). Consistently, mutations designed to only affect processing did not affect promoter activity (pink tones in Supplementary Fig. 4c). The ΔtRNA^Pro showed a very strong decrease in promoter activity, and this effect was further enhanced in the ΔC55A, ΔT54C/A58C, and ΔC55G variants (Supplementary Fig. 4c). Functional gRNA assays revealed a minor decrease in activity for double mutants affecting processing, consistent with their decreased 3′ processing activity (Supplementary Fig. 4d). Double-mutations designed to reduce promoter activity showed

an intermediate decrease in functional gRNA activity, consistent with the combination of their decreased promoter strength and partial loss of processing (Supplementary Fig. 4d). The ΔtRNA^Pro scaffold showed a strong loss of background gRNA activity, and the addition of other candidate mutations from our screen completely abrogated this leakiness in three out of four tested combinations (ΔC55A, ΔT54C/A58C and ΔC55G, Supplementary Fig. 4d). These results validate the predictions of our screen, but also suggest that additional synergistic effects may be possible by combining multiple mutations.

Having identified a number of tRNA variants that are potentially competent for gRNA excision from Pol-II transcripts

**Fig. 2** Mutagenesis screen identifies sequence determinants of tRNA processing and promoter activities. **a** Experimental design for a tRNA mutagenesis screen in human cells. Parallel libraries of partially degenerate tRNA[Pro] were created (dark orange = mutated positions), with and without a CMV promoter. Degenerate barcodes (light orange) were placed between the gRNA and tRNA sequences to allow variant identification following processing. A short buffer sequence was included between the barcode and the tRNA to protect the barcode from cleavage. RNA species from (+)CMV libraries were decapped to avoid the 5′cap-mediated inhibition of RNA circularization. **b**, **d**, **f** Heatmaps showing the effects of all single mutations at each nucleotide position on promoter activity ($n = 3$ paired pDNA/circRNA libraries with no Pol-II promoter) (**b**), 3′ processing ($n = 3$ paired pDNA/circRNA libraries with no Pol-II promoter) (**d**), and 5′ processing ($n = 3$ paired pDNA/circRNA libraries with CMV promoters) (**f**) (hatched squares = wild-type nucleotides; black squares = no measurement). In (**b**) promoter activity was calculated as the ratio of observed reads in the RNA fraction of a given mutation compared to its expected frequency in the library from sequencing the plasmid DNA (green borders = changes from wild type with a probability of 80% or greater, BEST test; only mutations with observations in all three libraries were included in significance testing). In **d**, **f** processing efficiency was inferred by averaging the binomial probability distributions (processed of total reads) across replicates then taking the point of highest probability as the final value estimate (green borders = changes from wild type with probability densities overlapping by less than 5%). **c**, **e**, **g** Corresponding tRNA diagrams showing for each modified position, the mutation which rendered the lowest (left) and highest (right) levels of their respective measurements (colors correspond to the heatmaps in (**b**), (**d**), (**f**), respectively). **h** Correlation plots for all combinations of measurements. Specific nucleotide changes are as indicated. Red lines reflect a linear fit. Source data are provided as a Source Data file

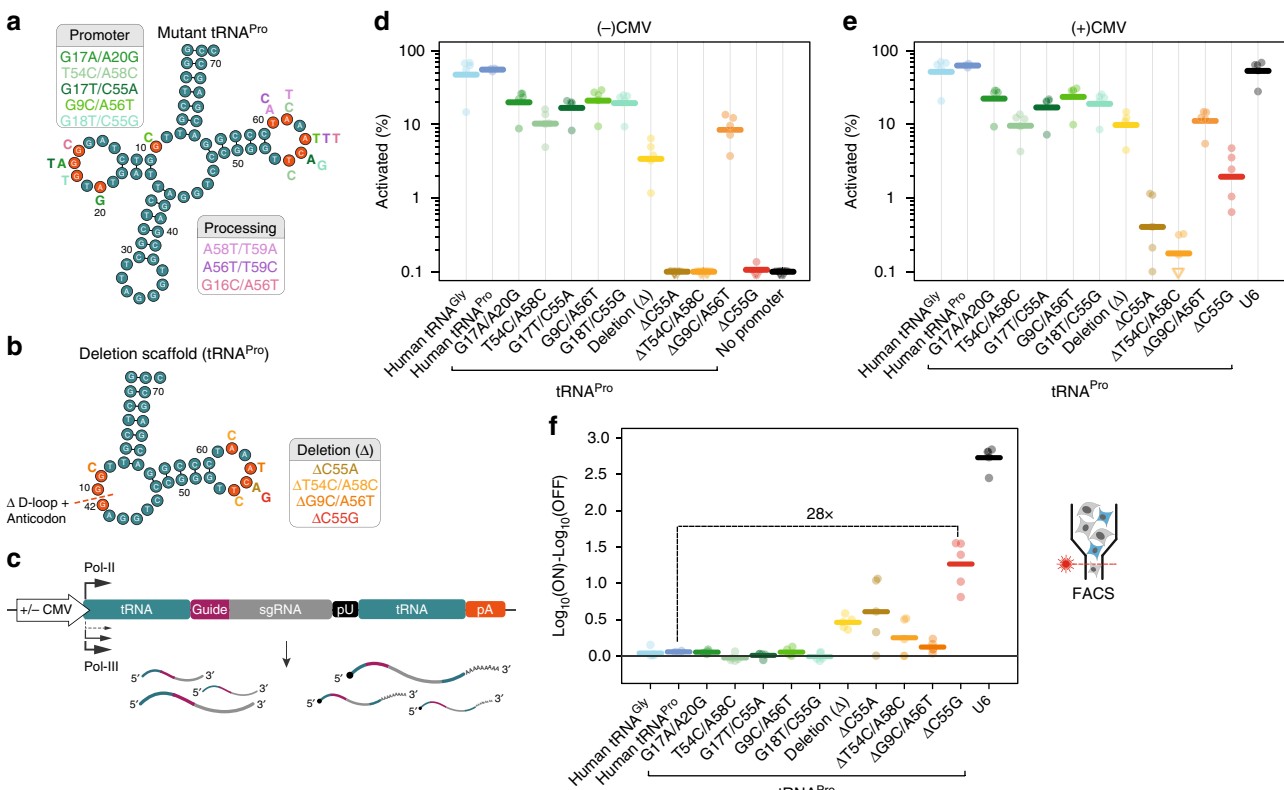

**Fig. 3** Combinatorial tRNA deletions and point mutations improve the ON−OFF specificity of Pol-II gRNA expression. **a**, **b** tRNA diagrams with mutation pairs predicted from sequencing data to affect 3′ processing only (pink tones), or promoter activity with minimal effect on processing (green/orange/red tones) (left panel). Selected pairs of mutations are shown as letters next to the wild-type (**a**) and ΔtRNA[Pro] (**b**) scaffolds. **c** Constructs used to test ON/OFF ratios of each variant combination in the presence and absence of a Pol-II promoter. **d**, **e** Percentage reporter ECFP[+] cells within transfected cells with (**d**) and without (**e**) a CMV promoter. Thick lines indicate geometric mean values. Each point represents an independent experiment ($n = 4–5$). **f** $\mathrm{Log_{10}}$(% ECFP[+] cells) in the ON condition compared to OFF condition for each tRNA and U6 (relative to no promoter) (thick lines = mean values; $n = 3–5$ independent experiments). The U6 and no promoter control are shared with Fig. 1 and Supplementary Fig. 4d. ΔC55G had a >99.9% probability of decreased activity compared to U6, but a 99% probability of increase compared to parental tRNA (paired BEST tests). Source data are provided as a Source Data file

and display reduced or no background activity, we next created paired constructs whereby gRNAs were flanked by engineered tRNAs (Fig. 3c, Supplementary Fig. 5a). We then measured the aggregate processing ability by circularization in the presence of a CMV promoter (ON-state). This revealed only a slight decrease in overall 3′+5′ processing compared to 3′ processing alone, in

both wild-type tRNA and double mutants, suggesting that the 5′ processing activity was not substantially impaired (Supplementary Fig. 1, 5b). In contrast, the ΔtRNA[Pro] scaffold showed significantly decreased processing in this assay suggesting that 5′ processing is impaired when the D-loop is entirely removed. Introducing other selected mutations further decreased

processing, as expected from their effects on 3′ processing. Several combinations, however, retained a readily detectable degree of processing. In particular, the ΔC55G tRNA[Pro] displayed the highest processing ability (Supplementary Fig. 5b) among combinations devoid of leakiness (Fig. 3d and Supplementary Fig. 4d). Quantification of gRNA levels showed strong ON−OFF ratios in all ΔtRNA[Pro] scaffold combinations (Supplementary Fig. 5c). Interestingly, RNA levels did not increase when a Pol-II promoter was added in front of the wild-type tRNA (Supplementary Fig. 5c) consistent with reports that active Pol-III promoters may inhibit nearby Pol-II activity[20,21].

Next, we tested the levels of functional gRNAs in each case and determined the reporter activation in the ON and OFF states. This analysis revealed poor ON−OFF ratios for wild-type tRNAs as well as the double mutants (Fig. 3d−f), as predicted by their high background expression and negligible increase in RNA abundance in the presence of a Pol-II promoter. In contrast, the ΔtRNA[Pro] scaffold and derivatives showed substantially improved ON−OFF ratios due to decreased or absent background activation (Fig. 3d). Importantly and consistent with our promoter and processing assays, while still lower than U6, the ΔC55G tRNA[Pro] had an ON−OFF ratio over an order of magnitude higher than the wild-type tRNA[Pro] (Fig. 3f).

To establish whether our tRNA deletion/mutant framework is generalizable, we introduced our top performing ΔC55G modification in a human tRNA[Gly] backbone (GCC; tRNAscan-SE ID: chr1.trna34). This analysis revealed similar elimination of background activity, and improved ON−OFF ratios as observed with the ΔC55G tRNA[Pro] (Supplementary Fig. 6). These results suggest that the principles described here can be applied to other tRNAs, which in combination could decrease the risk of recombination for multiplexed gRNA frameworks.

**Targeting endogenous genes with Pol-II-driven gRNAs**. We next sought to confirm that our engineered tRNA scaffolds are capable of enabling activation of endogenous genes. To this end, we used the synergistic activation mediator (SAM) system[22] together with varying combinations of our best performing tRNA[Pro] scaffolds (Δ, ΔC55A, and ΔC55G) to activate endogenous nerve growth factor receptor expression (NGFR, NM_002507). NGFR is a receptor tyrosine kinase involved in a variety of cell growth and differentiation processes[23,24]. As expected, in the absence of an active Pol-II promoter, NGFR levels were very high when using the wild-type tRNA[Pro] (paired, two-sided, BEST test > 99.9% probability > scrambled control). In contrast, various engineered tRNA scaffold combinations displayed NGFR levels that were near to, albeit slightly higher than, the scrambled gRNA control which had been used to set the background expression threshold (Fig. 4a and Supplementary Fig. 2e, f). In the presence of a Pol-II promoter, we observed significant increases in surface NGFR relative to control conditions ((−) CMV) for all combinations with the exception of wild-type tRNA[Pro] (paired, two-sided, BEST tests probabilities ≥99, <50% for wild type, Fig. 4b). The lower ON/OFF ratios compared to those observed in the reporter system are likely due to the presence of some background staining in these cells, thus reducing the sensitivity of detection. Overall, however, these results demonstrate that endogenous activation can be achieved using various combinations of engineered tRNA scaffolds.

Next, we tested the same combinations of tRNA scaffolds for their ability to mediate editing of an endogenous genomic locus using a standard T7 endonuclease assay. As a proof of concept, we used a gRNA targeting the 3′ UTR of programmed death-ligand 1 (PDL1)[25]. In the absence of a Pol-II promoter we observed negligible editing from most of our tested candidates, with only

the wild type showing strong and Δ/Δ showing weak, nonspecific editing above background (paired, two-sided, BEST test probability above background 97% and 77% respectively; Fig. 4c, d and Supplementary Fig. 7). In contrast, strong editing was observed for all engineered scaffolds in the presence of the CMV promoter with up to 10% indel formation at the targeted locus after 4 days (paired, two-sided, BEST test probabilities 91–96%) (Fig. 4c, d and Supplementary Fig. 7). As expected, the wild-type tRNA[Pro] was only marginally affected by the addition of the CMV promoter.

To determine the impact of Pol-II promoter strength on functional gRNA production, we also tested the PGK promoter, which has been reported to display ~0.25× the expression strength of CMV in HEK293T cells[26]. As expected, the editing efficiency was markedly reduced (~1% indel formation after 4 days) although it remained detectable in some experiments (Fig. 4c) and tended to increase beyond no promoter controls overall (paired, two-sided, BEST test probabilities 71–80%; Fig. 4c, d). It should be noted that these results were partially confounded by the limit of detection of the T7 endonuclease assay which is near the expected (and observed) output value for PGK at 4 days. Furthermore, the actual editing efficiency may be somewhat higher than observed due to the fact that antibiotic selection for Cas9 expressing cells was only 80% or less (Supplementary Fig. 2f).

Interestingly, the observed editing efficiency of ΔC55G/ΔC55G under the CMV promoter equates to 1–3% of cells each day producing sufficient levels of active gRNAs. This result fits well with the measurements of functional gRNA expression per day from our reporter assays (Fig. 3e). Considering that editing is a cumulative event and under the assumption that the rate remains constant, we modeled the frequency of edited cells over time (Supplementary Fig. 8). These analyses suggest that over physiologically relevant time scales editing could approach completion when a strong promoter is used together with our engineered tRNA scaffolds (1–3% editing/day). Conversely, according to this model, the mutation frequency would not reach saturation if weaker promoters were used (0.25% editing/day).

**Comparative analysis of existing Pol-II gRNA release systems**. Finally, we sought to benchmark our engineered tRNA scaffold against other systems previously employed for Pol-II transcribed gRNA excision[5–7,27]. As reported, all these systems were devoid of significant background activity (Fig. 5a). However, in our hands, most of these platforms displayed minimal gRNA-mediated transcriptional activation (Fig. 5b, c). The Csy4 protein/target combinations proved the exception, with consistent guide release when the full Csy4 target sequence was used to flank the guide (Fig. 5b, c). Consistent with previous reports, a minimal Csy4 target[27] displayed similar ON/OFF ratios to the full target sequence regardless of whether these sequences were in a transcript on their own or in the 3′ untranslated region (UTR) of a protein coding transcript. Interestingly, however, the recently reported Csy4[Nano] sequence[19] outperformed both the full and minimal Csy4 target variants (94.3–99.7% probability in paired, two-sided, BEST tests). In comparison to the ΔC55G tRNA[Pro], full and minimal Csy4 constructs were slightly less efficient in producing functional gRNAs (92.6–99.7% in paired, two-sided, BEST tests), while Csy4[Nano] was equivalent. These results both identify an optimal target sequence for Csy4-mediated Pol-II guide release, and demonstrate that our engineered tRNA scaffold performs better than most competitors including the native Csy4 target sequence. Importantly, the lack of activation observed in the absence of Csy4 protein and the negligible activation in the minimal poly-A terminator condition highlight the need of processing for functional guide production, a requirement that has been contested in some publications[5,6].

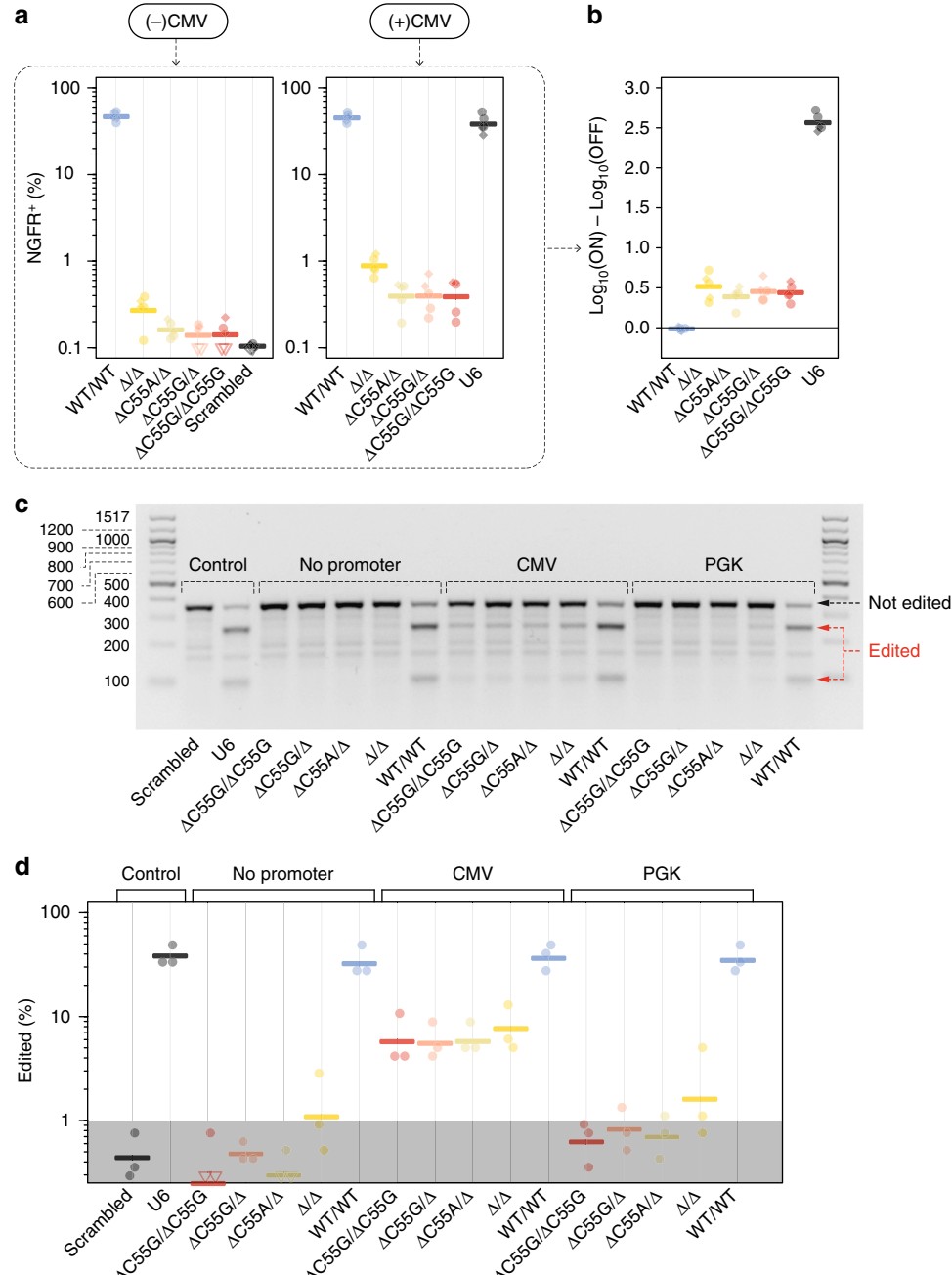

**Fig. 4** Engineered tRNA$^{Pro}$ scaffolds enable Pol-II-specific endogenous gene targeting. **a** Percentage of NGFR$^+$ cells within transfected cells for the OFF state ((−) CMV promoter) and the ON-state ((+) CMV promoter). Thick lines denote geometric mean values ($n = 5$ independent experiments; diamonds = day 4 with antibiotic selection; circles = day 3 no antibiotic selection; see Methods). Points at or below limit of detection are displayed as hollow triangles or diamonds for antibiotic selected experiments. Threshold of detection was set as containing ~0.1% of events in the scrambled gRNA control. **b** Log$_{10}$(% NGFR$^+$ cells) in the ON condition ((+) CMV promoter) compared to OFF condition ((−) CMV promoter). **c** Representative T7 endonuclease assay gel. tRNA scaffolds are shown beneath each plot with the 5′ variant/3′ variant indicated. A U6 promoter with a scrambled gRNA or with the targeting gRNA are included in as controls. Pol-II promoters for each set are noted above their respective lanes. Size in base pairs (bp) are shown next to the left side ladder. Black arrow = the location of uncut (not edited) amplicons; red arrows = digested (edited) amplicons. **d** Quantification of editing events from theT7 endonuclease gels. Thick lines represent geometric mean values ($n = 3$ independent experiments). Shaded area represents the 75% credible mass for the scrambled gRNA control (see also Supplementary Figure 7). Source data are provided as a Source Data file

## Discussion

In this study, we have developed a high-throughput method for screening tRNA functional parameters, and used it to identify the base dependencies of human tRNAs promoter activity, 3′ processing and 5′ processing. This information enabled us to rationally engineer a tRNA scaffold with substantially improved specificity for Cas9 gRNA expression from Pol-II promoters. This framework overcomes the limitations of previous tRNA-mediated release systems, which were compatible with multiplex gRNA delivery but not with spatial/temporal control of gRNA expression, due to their intrinsic Pol-III promoter activity.

To demonstrate the utility of this system in mammalian cells, we carried out transcriptional activation and editing of endogenous genes. All our engineered tRNA scaffolds induced

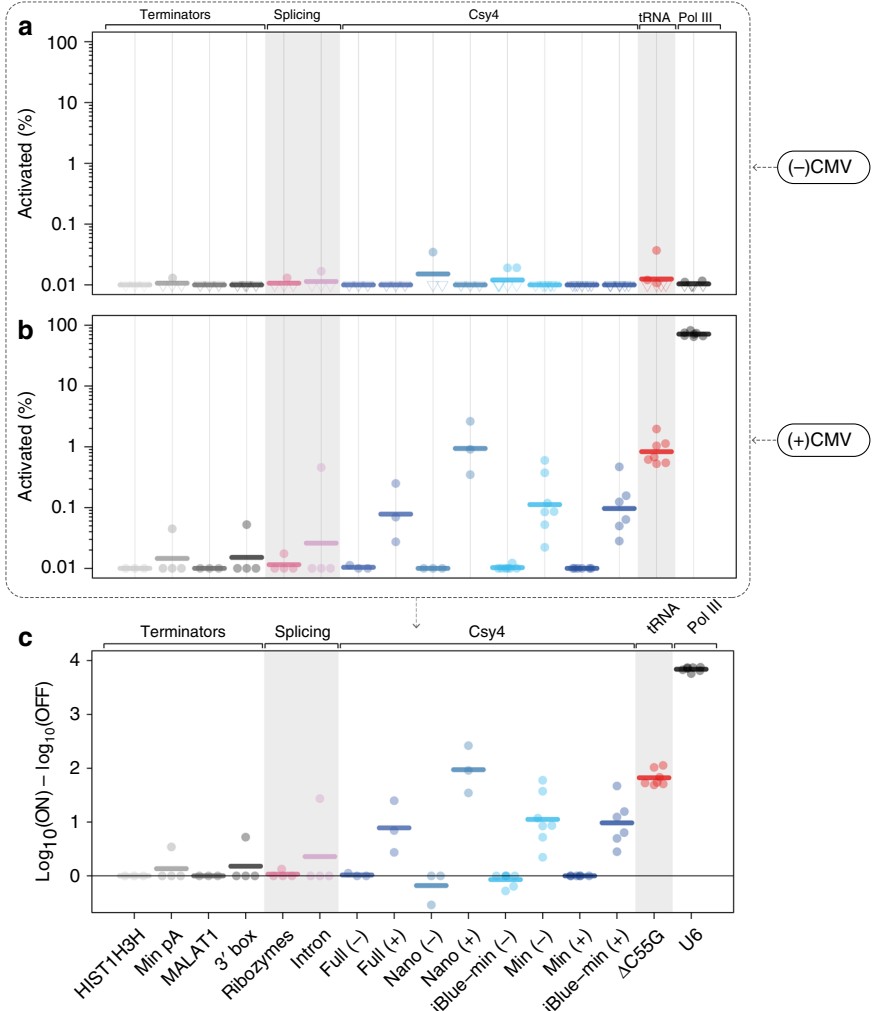

**Fig. 5** Comparison of top performing tRNA^Pro ΔC55G engineered variant with other reported systems for specific Pol-II sgRNA expression. **a**, **b** Percent reporter ECFP+ cells within transfected cells for the OFF state (no CMV promoter) (**a**) and the ON-state (with CMV promoter) (**b**). Points at or below the limit of detection are shown as hollow downward triangles. **c** $Log_{10}$(% ECFP+ cells) in the ON condition (with CMV promoter) compared to OFF condition (no CMV promoter). Thick lines represent geometric mean values ($n = 3-6$ independent experiments). Terminators refer to the use of alternative termination sequences to the standard poly A and include the histone 1h3h terminator, a minimal poly A sequence[5], the MALAT 1 terminator[5], as well as the U1 snoRNA 3′ box[5]. The ribozyme scaffold contains a gRNA flanked by hammerhead and HDV ribozymes[7]. The intron system contains a gRNA embedded within an artificial intron inside the mKate fluorphore[6]. Csy4 variants include the full Csy4 target sequence[4], the Csy4^Nano target from[19], and minimal Csy4 targets[27] either on its own in a transcript, or in the 3′ UTR of an iBlue transcript. (+) indicates that the Csy4 protein is present, and (−) indicates that it is absent. Source data are provided as a Source Data file

significant Pol-II-dependent activation of NGFR, albeit in a minority of cells. Even under these conditions, the system will enable various applications, including tissue-specific paracrine factor expression where even few cells producing high levels could have widespread effects. We also showed that Pol-II-mediated editing of endogenous genomic loci is possible using our system. Although the absolute editing rate was relatively low, our data and mathematical model suggest that high levels of indel formation could be achieved within a physiological timeframe. This could facilitate induction of genome editing events in multiple independent cell lineages.

Information from the two Pol-II promoters used in this study suggests that promoter strength has a substantial impact on the indel formation. Processing ability was also observed to be an important predictor of functional gRNA production. Based on these observations, we propose that the rate of editing will be a function of both promotor strength and processing efficiency.

Further research into the effects of multiple mutations/deletions in human tRNAs may improve the processing activity without loss of specificity, and thus could improve editing rates and allow weaker promoters to be used.

A direct comparison of existing Pol-II guide RNA release systems allowed us to both demonstrate the superior performance of our engineered tRNA scaffold and identify the optimal target sequence for the best alternative platform using the Csy4 endonuclease. These two systems showed good ON/OFF ratios and clean background profiles. The engineered tRNA scaffold, however, has several advantages, including generalizability across species, decreased recombination risk when combining multiple tRNA backbones, and is devoid of any exogenous proteins, thus reducing potential immunogenicity and system complexity. Nonetheless, both systems will likely prove useful in various experimental setups, and may behave differently in response to local RNA context and promoter strength.

Overall, our findings provide insights into the functional characteristics of human tRNAs and advance existing tools both for the study of tRNA function and for inducible Cas9 gRNA expression, thus enabling the implementation of more complex research applications.

## Methods

**Cloning and construct assembly**. All restriction enzyme digestions were performed in suggested buffers either as double digestions (if compatible) or sequential digestions as appropriate. In the case of sequential digestions, a Qiaquick PCR purification (Qiagen) with 30 µl elution volume (Buffer EB) was used to re-purify between digestions. Following digestion, vectors were treated with 5 U of Antarctic Phosphatase for 30 min at 37 °C (NEB). All vectors were then gel purified using a Qiagen gel extraction kit as per the manufacturer's instructions but with Qiagen MinElute PCR purification columns substituted for Qiaquick columns and a final elution volume of 15 µl buffer EB. Inserts were purified prior to ligation with either a standard Qiagen gel extraction protocol, or a Qiaquick PCR purification (Qiagen) as appropriate. All standard ligations were performed using 200 or 400 U T4 DNA ligase in T4 DNA ligase buffer (NEB) with approximate insert to vector ratios between 1:1 and 1:10. Ligation reactions were incubated at room temperature for 5–30 min (5–10 min for single insert ligations, 20–30 min for ligations with more inserts). Ligase was then heat inactivated by 10-min incubation at 80 °C. Ligation reactions were then cooled on ice for 2 min, then 1–3 µl added to 10–50 µl Subcloning Efficiency™ DH5α™ Competent Cells (Thermo Fisher Scientific) with a maximum ligation to bacteria ratio of 1:10. Cells were then incubated on ice for 15–30 min, heat shocked at 42 °C for 20 s in a water bath, then 200–500 µl S.O.C. medium (homemade) added. Cells were then incubated for 40 min to 1 h at 37 °C with shaking and 200 µl plated onto LB agar plates with 100 µg/ml ampicillin. Plates were incubated at 37 °C overnight. Individual colonies were then picked into 5–7 ml LB + 100 µg/ml ampicillin (Sigma) and incubated again overnight at 37 °C with shaking. Plasmids were then purified using Qiaprep Spin Miniprep columns (Qiagen). Finally, plasmids were verified by appropriate diagnostic digests and sequencing with appropriate primers (generally one of pBR322_ori-F: CACC TCTGACTTGAGCGTCG, AmpR-R: GGTTATTGTCTCATGAGCGG, SV40po-lyA-R: TACTCGAGGGATCCTTATCGATTTTACC, or forUAS-F: CCAATCTC GAGGAGGCTAGGGATGAAGAATAAAAG) using Eurofins Genomics sequencing service.

**PCR amplifications for cloning**. For all PCR amplifications used in cloning, Phusion® High-Fidelity PCR Master Mix with GC Buffer (New England BioLabs, NEB) was used for amplification with 500 nM of each primer (primers are as indicated for individual reactions). Amplification conditions were as follows: 98 °C for 60 s, 40 cycles of 98 °C for 10s, optimal annealing conditions as determined by the NEB Tm calculator for 30 s, 72 °C for 10-60 s, then 5 min at 72 °C.

**Oligonucleotide phosphorylation and annealing**. For all oligo annealing reactions, 10 µl containing 10 µM each of the forward and reverse oligos, 1× T4 DNA ligase buffer, and five units of T4 Polynucleotide Kinase (NEB) were incubated at 37 °C for 30 min, ramped to 95 °C for 5 min then cooled to 25 °C at a rate of 0.1 °C/s. The annealed oligos were then used as inserts for cloning reactions where applicable.

**Backbone creation**. An initial vector was created by three-insert restriction cloning using a backbone containing a pBR322 origin with ampicillin resistance and adjacent SphI and SbfI restriction sites. To generate the first insert, we first amplified the poly-A signal and pause site out of the 8xCTS2-MLP-EYFP construct (Addgene plasmid # 55197, a gift from Timothy Lu, ref. [19]) (Oligo Pair 1, Supplementary Table 1). The amplified fragment was purified by Qiaquick PCR purification (Qiagen) and incubated with five units of standard Taq polymerase in standard Taq buffer with 2.5 mM dATP (NEB) for 15 min at 72 °C to add A overhangs. This was then TA cloned using a TOPO® TA Cloning® Kit (Thermo Fisher Scientific). Resulting clones were then reamplified (Oligo Pair 2, Supplementary Table 1) and the resulting product digested with SphI and KpnI to obtain the first insert. A modular scaffold containing multiple cloning sites and a minimal adenovirus major late promoter (MLP) was ordered from IDT as a gBlock (Supplementary Note). This was amplified (Oligo Pair 3, Supplementary Table 1), A overhangs added as for insert one, and TA cloned using a TOPO® TA Cloning® Kit (Thermo Fisher Scientific). Resultant TOPO clones were digested with KpnI and FseI to obtain the middle insert. The final insert, the SV40 polyA sequence, was amplified from the 8xCTS2-MLP-ECFP construct (Addgene plasmid # 55198, a gift from Timothy Lu, ref. [19]) (Oligo Pair 4, Supplementary Table 1) followed by digestion with FseI and SbfI-HiFi (both from NEB). This resulted in the minimal backbone used for downstream cloning, Backbone 1. Finally, to allow gating for cells which received the gRNA plasmids, the SV40 promoter-iBlue-SV40 polyA site was amplified out of the U6-CTS2 construct (ref. [19]) (Oligo Pair 5, Supplementary Table 1), digested with XhoI and AatII and inserted into the same sites in Backbone 1 to generate Backbone 2.

**8xCTS2 reporter modification**. To allow the functional analyses to be performed only on cells which had received all necessary plasmids, we inserted an SV40 promoter-mCherry-SV40 polyA cassette into the backbone of the 8xCTS2-MLP-ECFP construct between the PciI and SalI sites. The insert was amplified from an available plasmid using primers which added the PciI and SalI sites respectively (Oligo Pair 6, Supplementary Table 1).

**tRNA promoter testing**. To create the constructs used in testing the Pol-III promoter activity of tRNA, 3–4 insert standard cloning was performed. The tRNAs comprised 1 or 2 pairs of (separately) annealed oligos as listed in Supplemental Table 2. In cases with a mutation in one part but not the other, the wild-type oligo pair was used for the opposite side insert. The CTS2 guide sequence was another pair of annealed oligos (Supplemental Table 2). All annealed oligo pairs had unique 4 bp overhangs on each side to allow scar-free ligation to their partners. Finally, the single guide RNA (sgRNA) was amplified from pX330-U6-Chimeric_BB-CBh-hSpCas9 (Addgene plasmid # 42230, a gift from Feng Zhang) using Oligo Pair 7 (Supplementary Table 1), followed by a sequential digest with BpiI (Thermo Fisher Scientific) and FseI (NEB). These inserts were placed into Backbone 2 between the SacI and FseI sites (NEB). In some cases, the minimal CMV was then replaced using AvrII and SacI-HF (both from NEB) with the annealed oligo set "noPromoter". This was not done for the D-loop anticodon-deleted tRNA^Pro and the ΔC55G tRNA^Pro, as in functional tests it was not shown to make a difference.

**Pol-II construct cloning**. Cloning for constructs to test the effects of Pol-II promoters was done in two ((−)CMV) or three ((+)CMV) stages. First, the existing tRNA variant was amplified using appropriate primers (Supplemental Table 3) with an added FseI restriction site on each side. This product was then cloned into the FseI site of the parental plasmid (containing a tRNA-CTS2 sgRNA) located between the sgRNA and SV40 polyA site. Following screening for correct orientation (by sequencing), the SV40 promoter-iBlue-SV40 polyA cassette was removed by AatII + XhoI digestion (NEB) and replaced with the annealed oligo set "iBlueRemover". This was done in all constructs aside from the D-loop/Anticodon deleted tRNA^Gly and the ΔC55G tRNA^Gly as in initial tests Pol-II specific activation was anticorrelated with iBlue levels, suggesting possible crosstalk with the Pol-II tRNA cassette, although the functional activation levels achieved ignoring the iBlue did not appear to be affected. This step yielded the (−)CMV constructs. Finally, a CMV promoter amplified from a microRNA reporter plasmid (Michaels et al. Nat. Commun., in press, Oligo Pair 8, Supplementary Table 1) was inserted into the AscI-SacI site in the (−)CMV construct (replacing the minimal CMV or "noPromoter" region).

**Alternative processing strategies**. For the intronic guide constructs, we ordered a gBlock (Supplementary Notes) containing mKate with an intronic sgRNA backbone and a cloning site to insert the gRNA sequence based on ref. [6]. This was amplified to add in restriction sites for cloning into Backbone 1 (Oligo Pair 9, Supplementary Table 1). This was then cloned between the SacI and FseI sites of Backbone 1 to give an intermediate construct for the (−)CMV version. Simultaneously, the same CMV promoter amplicon from the Pol-II constructs was co-inserted into Backbone 1 together with the mKate intronic gRNA construct between the AscI and FseI sites of Backbone 1 to give the (+)CMV intermediate. Annealed oligos ("intronic CTS2") were then inserted into the gRNA cloning site of each of these to clone the final intronic gRNA constructs.

To clone the ribozyme release system, a three-insert cloning between the SacI and FseI sites of the (+)CMV and (−)CMV intronic gRNA constructs was performed (removing the complete mKate/intronic guide). The first insert was a set of annealed oligos, "HH_CTS2", containing the hammerhead ribozyme from ref. [7] specific for CTS2 (the first six bases of the ribozyme complementary to the first six bases of the guide) with the CTS2 guide. This pair contained 4 bp overhangs compatible with SacI and the sgRNA. The second insert was the SAM-sgRNA with a BsmBI cloning placeholder. This was amplified from Backbone 1 (Oligo Pair 10, Supplementary Table 1) and digested with BsmBI and BpiI to give compatible sticky ends. The final insert was the HDV ribozyme from ref. [7]. This ribozyme sequence was ordered as an oligo from IDT (Supplementary Notes) and amplified to add a BpiI site to the front such that the digested product had sticky ends compatible to the sgRNA and FseI site to the back (Oligo Pair 11, Supplementary Table 1). This was sequentially digested with BpiI and FseI to give the final fragment.

For all alternative transcriptional terminators, the (+)CMV intronic gRNA construct was used as a vector. Inserts were then placed between the AgeI and XhoI site for the (+)CMV terminator constructs (leaving the initiator consensus of the CMV promoter immediately upstream of the guide while removing the SV40 polyA), and between the AscI and XhoI sites for the (−)CMV terminator constructs (removing the CMV and SV40 poly A entirely). In all cases this was done by two-insert cloning. The first insert contained the gRNA amplified from the ribozyme construct using primers which added BpiI sites on either end. For the (−)CMV constructs the BpiI site in the forward primer (Oligo Pair 12, Supplementary Table 1) created a sticky end compatible with AscI, and for the (+)CMV constructs the BpiI site in the forward primer (Oligo Pair 13, Supplementary Table 1) created a sticky end compatible with AgeI. In both cases these shared a common reverse

primer. The second insert contained the terminators. This was made up of annealed oligos with 4 bp overhangs compatible with the gRNA at the 5′ end and with the vector *Xho*I site on the 3′ end for the minimal polyA and HIST1H3H terminators. For the MALAT1 and 3′ Box terminators the sequences were ordered from IDT (Supplementary Notes) then amplified with *Bpi*I flanked primers to yield compatible sticky ends (MALAT1: Oligo Pair 14, 3′ Box: Oligo Pair 15, Supplementary Table 1).

The basic Csy4 constructs were cloned as a single insert into the *Sac*I *Fse*I site of the intronic gRNA constructs. The insert containing the gRNA was amplified from the HIST1H3 terminator construct (Full and Minimal) or *Backbone 1* (for Nano) using primers which added the full (GTTCACTGCCGTATAGGCAGCTAAG AAA), Nano[19] (ACTACCGTATAGGCAGC), or minimal (CTGCCGTATAGGC AGC) Csy4 target sequence and *Sac*I and *Fse*I sites respectively (primers in Supplementary Table 3). The Csy4[Nano] guide had a *Bsm*BI placeholder in place of a spacer, which was then replaced with the CTS2 spacer by single insert cloning using annealed oligos (Supplementary Table 2). The iBlue-Csy4-gRNA constructs were made in a two-insert cloning between the *Asc*I and *Fse*I sites of the intronic guide constructs. The first insert was amplified from a microRNA reporter plasmid (Michaels et al. *Nat. Commun.*, in press) and contained either the iBlue sequence alone (Oligo Pair 16, Supplementary Table 1), or a CMV promoter followed by iBlue (Oligo Pair 17, Supplementary Table 1). In both cases these were flanked by *Asc*I sites and *Sac*I sites to make the inserts compatible with the vector and same gRNA amplicon used for the minimal Csy4 constructs (the second insert).

**tRNA screening constructs.** Cloning for the single nucleotide variant tRNA libraries was performed in three (−)CMV or four (+)CMV stages. For the (+) CMV constructs, first the CMV promoter amplicon used for the Pol-II constructs was inserted between the *Asc*I and *Sac*I sites of one of the Pol-III tRNA constructs (tRNA-pol-III terminator, SV40 promoter-iBlue-SV40 polyA). The next stage (shared between the (+)CMV and (−)CMV libraries) involved a three-insert cloning. The first insert comprised a CTS2-sgRNA amplified from the Pol-III tRNA constructs with primers adding a *Sac*I site to the 5′ end and a *Bpi*I site to the 3′ end such that the overhang was complementary to the middle insert (Oligo Pair 18, Supplementary Table 1). The second insert was a pair of annealed oligos containing a pair of outward facing *Bpi*I sites to allow downstream GoldenGate assembly ("BpiI_placeholder", Supplementary Table 2). This annealed oligo had 4 bp overhangs complementary to the overhangs created by the *Bpi*I digestions of the inserts on either side. The final insert was also a CTS2-sgRNA amplified from the Pol-III tRNA constructs but with primers now adding a *Bpi*I site to the 5′ end, retaining the pol-III terminator, and adding an *Fse*I site to the 3′ end (Oligo Pair 19, Supplementary Table 1). Sequencing validation of the (−)CMV construct at this stage revealed a recombination event between the SV40 polyA downstream of the main construct and downstream of the iBlue (which is in the reverse orientation). This retained bidirectional polyA sites thus still allowing proper iBlue transcript formation, and the event does not affect the Pol-III transcribed region. Therefore, this construct was still retained for downstream cloning.

We next added in the flanking barcode sequences and buffers with a paired *Bsm*BI site between them by *Bpi*I-based GoldenGate assembly. To do so, we ordered a 116 bp barcode library ultramer from IDT ("Barcode library" sequence, Supplementary Notes). This was amplified in three independent PCR reactions with 100 fmol template per reaction (Oligo Pair 20, Supplementary Table 1). Following amplification products were gel purified using Qiagen gel extraction kit as per the manufacturer's instructions but with Qiagen MinElute PCR purification columns. These were then combined at equimolar ratios to give the final insert. This was done to minimize potential PCR error and sampling biases[28,29]. A total of 20 GoldenGate assembly reactions were then assembled and cycled as follows. The reaction was composed of 200 ng of vector, 16.5 ng of Barcode Library, 2 µl of 10x T4 DNA Ligase Buffer (NEB), 0.5 µl of T4 DNA Ligase (NEB), 0.5 µl of BpiI (10 U/µl, Thermo Fisher Scientific) and H$_2$O up to 20 µl. Reaction conditions were 30 cycles of 37 °C for 5 min, 16 °C for 5 min, then 50 °C for 5 min, 80 °C for 5 min.

Following cycling all 20 GoldenGate reactions were pooled and copurified by Qiaquick PCR purification (Qiagen) with a 30 µl elution volume. A total of three transformations were performed. For these, 5 µl of the purified GoldenGate reaction was then added to 30 µl of NEB 10-beta Competent *E. coli* (NEB) and transferred to an electroporation cuvette with a 1 mm gap (VWR). This was then electroporated on an Eppendorf Eporator® with 1.6 kV over 5 ms. Volume was then immediately topped up to 1 ml with 37 °C SOC (homemade) and transferred to a 1.5 ml microtube and incubated at 37 °C with shaking for 1 h. All three transformations were then combined and the total volume brought to 8 ml with warm SOC. This was used to inoculate two 245 mm dishes (Corning). 0.1 µl was also plated onto a 10 cm dish to allow an estimation of colony number. These were incubated for 16 h at 32 °C. Following incubation colonies were harvested by scraping using bacteria spreaders with LB washes. Harvested colonies were collected into a 50 ml Falcon tube (Thermo Fisher Scientific), spun down at 3000 × *g*, and the supernatant removed. Pellets were then weighed and split across a multiple Qiagen plasmid Midiprep columns and plasmids purified as per the manufacturer's protocols.

For the final stage of library cloning 36 µg of each vector from the previous stage was digested in a 360 µl reaction with a total of 120 U of *Bsm*BI in NEBuffer 3.1 (both from NEB) for 2 h at 55 °C. Following digestion, 60 units of Antarctic

Phosphatase and associated buffer was added to the reaction and incubated for an additional 30 min at 37 °C. Digestions were then gel purified with a Qiagen gel extraction kit as per the manufacturer's instructions. Inserts were prepared by triplicate PCR amplification (Oligo Pair 21, Supplementary Table 1) from the tRNA variant library oligo (Supplementary Notes), gel purification, and equimolar pooling as was done for the Barcode library insert. A total of 20 standard ligation reactions were then set up as follows. The reaction was composed of 500 ng of vector, 24.74 ng of tRNA variant library, 2 µl of 10x T4 DNA Ligase Buffer, 0.5 µl of T4 DNA ligase and H$_2$O up to 20 µl.

These reactions were incubated at 16 °C for 16 h. These were then pooled, copurified, transformed, plated, harvested and final plasmids extracted by Qiagen plasmid Midiprep exactly as was done in the previous stage.

**tRNA/guide cloning for endogenous gene targeting.** To efficiently assemble varying tRNA flanked guides, we first prepared a GoldenGate-compatible backbone using the 8xCTS1-MLP-EYFP construct as a starting point. As a first step we removed the *Bpi*I recognition site in the backbone by digesting with *Pci*I and *Bpi*I, and replaced with an annealed oligo pair (Oligo Pair 22, Supplementary Table 1). We next removed the EYFP fragment by *Nco*I, *Fse*I digestion and replaced it with an annealed oligo pair containing a *Bpi*I placeholder to act as a GoldenGate acceptor (Oligo Pair 23, Supplementary Table 1). This plasmid "8xCTS1-GGacceptor" was then used as a backbone for further cloning.

Inserts were made by a combination of oligo annealing and PCR amplification. tRNA variants were amplified from earlier constructs containing a single copy of the tRNA using the following primer pairs: (*Bpi*I-CAAC-5p-tRNApro-F+*Bpi*I-GCCC-5p-tRNApro-R) for 5′ position deletion scaffold tRNA, (*Bpi*I-14bpBuf-3p-deltRNApro-F+3p-del-tRNApro-R) for 3′ position deletion scaffold tRNA, (*Bpi*I-CAAC-5p-WTtRNApro-F+*Bpi*I-GCCC-5p-tRNApro-R) for 5′ position wild-type tRNA, and (*Bpi*I-14bpBuf-3p-WTtRNApro-F+3p-del-tRNApro-R) for 3′ position wild-type tRNA (primer sequences in Supplementary Table 3). The basic sgRNA with a *Bsm*BI placeholder for subsequent spacer insertion assembled in two pairs of annealed oligos (Oligo Pair 24, Supplementary Table 1) and (Oligo Pair 25, Supplementary Table 1). Finally the SAM-guide[22] with a *Bsm*BI placeholder was amplified from *Backbone 1* (Oligo Pair 26, Supplementary Table 1).

GoldenGate reactions were then assembled in a combinatorial manner to generate all no promoter constructs with *Bsm*BI placeholders. The reaction was composed of 50 ng of vector, ~5 ng of 5′ tRNA, ~12 ng for SAM-sgRNA or 1 µl 1:50 sgRNA oligos, ~8 ng of 3′ tRNA, 1 µl of 10x T4 DNA Ligase Buffer (NEB), 0.25 µl of T4 DNA Ligase (NEB), 0.25 µl of BpiI (10 U/µl, Thermo Fisher Scientific) and H$_2$O up to 10 µl. Reaction conditions were 30 cycles of 37 °C for 5 min, 16 °C for 5 min, 37 °C for 30 min 50 °C for 5 min, 80 °C for 5 min.

All GoldenGate reactions were then transformed as per standard ligations above.

Spacers were then inserted as annealed oligos following *Bsm*BI digestion of the no promoter constructs, with the NGFR spacer (Oligo Pair 27, Supplementary Table 1) being inserted into those constructs with the SAM-sgRNA, and the PDL1 spacer (Oligo Pair 28, Supplementary Table 1) being inserted into those constructs with the standard sgRNA.

For insertion of the CMV promoter, it was first amplified from previous CMV constructs (Oligo Pair 29, Supplementary Table 1). The no promoter constructs were then digested with *Nhe*I and *Bsm*BI, thus removing the 8xCTS1 through to the sgRNA. The amplified CMV promoter and the 5′ tRNA (from the initial GoldenGate) were then both digested with *Bpi*I and inserted together with the final spacer (annealed oligos above) to the *Nhe*I/*Bsm*BI-digested no promoter backbone. Note, while it was possible to generate the CMV containing construct directly in the initial GoldenGate step, we found this to be very inefficient in this case. Alternatively, the promoter could also be changed by amplifying the CMV flanked on both sides by *Nhe*I sites directly; however, this would require additional screening for directionality.

Finally, to generate the PGK constructs, PGK was amplified from PGK1p-Csy4-pA (Construct 2) (Addgene plasmid # 55196, a gift from Timothy Lu, ref. [4]) (Oligo Pair 30, Supplementary Table 1). The amplified PGK was then digested with *Nhe*I and *Bpi*I. The CMV containing tRNA-flanked sgRNA constructs were then digested with *Nhe*I and *Age*I, and the digested PGK promoter inserted by standard ligation.

**Cas9 and MS2 variations cloning.** Cloning for dCas9-VP64-T2A-Puro and MS2-P65-HSF1-T2A-BleoR involved dCAS9_VP64_GFP, MS2-P65-HSF1_GFP, and pSpCas9(BB)-2A-Puro (PX459) V2.0 which were gifts from Feng Zhang (Addgene plasmid #s 61422, 61423, 62988). To generate the dCas9-VP64-T2A-Puro, the T2A-puromycin resistance gene was amplified from PX459 V2.0 (Oligo Pair 31, Supplementary Table 1), digested with *Nhe*I and *Eco*RI, and inserted between the *Nhe*I and *Eco*RI sites of the dCAS9-VP64_GFP plasmid, replacing the GFP. To construct the MS2-P65-HSF1-T2A-BleoR construct, the *Bleo*R gene was amplified from the dCAS9-VP64_GFP vector (Oligo Pair 32, Supplementary Table 1) and digested with *Bpi*I and *Eco*RI. A pair of oligos was then annealed to replace the T2A sequence (Oligo Pair 33, Supplementary Table 1). Both components were then inserted between the *Nhe*I and *Eco*RI site of the MS2-P65-HSF1_GFP, again replacing the GFP sequence. To generate versions of dCAS9-VP64_GFP and MS2-P65-HSF1_GFP without the GFP, these were each digested with *Nhe*I and *Eco*RI

and a pair of annealed oligos containing a stop codon put in instead (Oligo Pair 34, Supplementary Table 1), thus replacing the T2A-GFP. Finally, to generate a PX459 V2.0 without sgRNA, PX459 V2.0 was digested with *Pci*I and *Xba*I and this replaced with annealed oligos (Oligo Pair 35, Supplementary Table 1), thus removing the U6-sgRNA cassette.

**HEK-293T cell maintenance.** HEK-293T cells (purchased from ATCC, ATCC-CRL-11268) were cultured in Dulbecco's modified Eagle's medium (DMEM) supplemented with 10% fetal bovine serum (FBS, E.U.-approved, South America origin, both from Gibco). Cells were maintained at 37 °C 5% $CO_2$ and passaged every 2–4 days at a ratio of 1:3–1:10. Cells tested negative for mycoplasma at least every 6 months using either a VenorGeM® Mycoplasma Kit (Minerva Biolabs) according to the manufacturer's protocols, or using a set of primers from refs. [30,31] with Phusion® High-Fidelity PCR Master Mix with GC Buffer (NEB). Cycling conditions for mycoplasma testing were 98 °C for 60 s, 35 cycles of 98 °C for 30 s, 70 °C for 30 s, 72 °C for 30 s, then 5 min at 72 °C.

**Transfections and DNA and RNA extractions.** For HEK-293T cell transfections, cells were plated the night prior to transfection in 12-well tissue culture plates (Corning) in 500 μl DMEM + 10% FBS at numbers such that the next day they were at 60–80% confluence. Prior to transfection the media was removed and the wells washed once with 500 μl PBS. The media was changed to 450 μl DMEM + 2% FBS. During this time, sufficient plasmid DNA such that at final volume (500 μl for these experiments) each plasmid would have a final concentration of 100 pM (50–500 ng each, depending on plasmid size) was brought up to 50 μl in Opti-MEM™ (Gibco) containing 1.5 μg polyethylenimine (PEI, Sigma). The solution was then mixed vigorously by vortexing for 10 s, let stand for 15 min at room temperature, and added drop-wise to the cells. Cells were then incubated for 4 h at 37 °C 5% $CO_2$. Transfection media was then removed and replaced with fresh DMEM + 10% FBS and the cells left for an additional 20 h (24 h from the start of transfection). Wells were then harvested using 0.05% Trypsin-EDTA (Thermo Fisher Scientific). Following trypsinization, half of the total cells from each well were taken for flow cytometry. The remainder were spun down at 300 × *g* for 5 min, the supernatant removed, and the pellet snap frozen on dry ice. On thaw, RNA was extracted using ChargeSwitch® Total RNA Cell Kit (Thermo Fisher Scientific) as per the manufacturer's instructions using an equal volume of 60 °C elution buffer (E7) to input beads. For the tRNA screening library transfections, instead of ChargeSwitch® extraction, 99% of cells were coextracted for RNA and DNA using the AllPrep DNA/RNA/miRNA Universal Kit (Qiagen) as per the manufacturer's instructions. The remaining 1% was taken to check transfection efficiency by flow cytometry.

Experiments using the ΔtRNA$^{Gly}$ backbones (Supplementary Fig. 6) were transfected as follows. DNA was prepared such that the final concentration of each plasmid would be at a final concentration of 156 pM (200–750 ng DNA). This was then brought up to 50 μl in OptiMEM with 6 μg PEI, vortexed to mix and incubated for 15 min at room temperature. The DNA-OptiMEM-PEI mix was then added to 250,000 HEK-293T cells in suspension in 450 μl DMEM + 5% FBS and plated into a well of a 24-well plate. Plates were then harvested 20–24 h later for flow cytometry.

For endogenous gene activations 100,000 HEK293T cells were plated in a 24-well plate in 500 μl DMEM + 10% FBS 4 h prior to transfection. Immediately prior to transfection these were changed to 200 μl DMEM + 2% FBS. Sufficient plasmid DNA such that at final volume (250 μl for these experiments) each plasmid would have a final concentration of 100 pM (50–225 ng each, depending on plasmid size, total ~550 ng plasmid/well) was brought up to 50 μl in Opti-MEM™ (Gibco) containing 1.5 μg PEI (Sigma). Each cell received either dCas9-VP64-T2A-Puro (or dCas9-VP64 no GFP), MS2-P65-HSF1-T2A-BleoR (or MS2-P65-HSF1 no GFP), the tRNA flanked NGFR guide construct for testing, and piRFP670-N1[32] (Addgene plasmid # 45457, a gift from Vladislav Verkhusha) as a transfection control. The plasmid optiMEM solution was then mixed vigorously by vortexing for 10 s, let stand for 15 min at room temperature, and added drop-wise to the cells. Cells were then incubated for 4 h at 37 °C 5% $CO_2$. Transfection media was then removed and replaced with fresh DMEM + 10% FBS. In the no selection experiments (three experiments) cells were left for 3 days total prior to harvest. For experiments with drug selection (two experiments), the next morning wells were supplemented with 1 μg/ml Puromycin Dihydrochloride and 200 μg/ml Zeocin™ (both from Gibco), and these incubated for an additional 3 days (4 days total). At the end of these incubations, cells were harvested using 0.05% Trypsin-EDTA (Thermo Fisher Scientific), washed once in PBS + 2% FBS. For drug selected cells they were then resuspended in 50 μl of PBS + 2% FBS + 0.25 μl Zombie Red™ Fixable Viability Dye (Biolegend) and incubated on ice for 15 min in the dark. These were then washed once in 1 ml PBS + 2% FBS. All samples were then stained in 50 μl PBS + 2% FBS + 0.5 μl mouse anti-human NGFR aka CD271 (clone C40–1457) V450 (BD Biosciences, cat #: 562123) for 30 min on ice in the dark. Cells were washed once with 1 ml PBS + 2% FBS, filtered then analyzed by flow cytometry.

For *PDL1* editing 100,000 HEK293T cells were plated in 500 μl DMEM + 10% FBS 4 h prior to transfection. Immediately prior to transfection these were changed to 200 μl DMEM + 2% FBS. Sufficient plasmid DNA such that at final volume (250 μl for these experiments) each plasmid would have a final concentration of

250 pM (150–360 ng each, depending on plasmid size, ~500 ng total/well) was brought up to 50 μl in Opti-MEM™ (Gibco) containing 1.5 μg PEI (Sigma). In these tests, each well received an sgRNA cassette removed PX459 V2.0 along with the tRNA-flanked PDL1 guide construct for testing. The plasmid optiMEM solution was then mixed vigorously by vortexing for 10 s, let stand for 15 min at room temperature, and added drop-wise to the cells. Cells were then incubated for 4 h at 37 °C 5% $CO_2$. Transfection media was then removed and replaced with fresh DMEM + 10% FBS. The next morning wells were supplemented with 1 μg/ml Puromycin Dihydrochloride (Gibco) and the cells incubated for 3 additional days prior to harvest (4 days total). At harvest, cells were washed 2× with PBS and 100 μl lysis buffer added. Lysis buffer consisted of 0.5% Tween-20 (Sigma), 50 mM Tris-HCl (Invitrogen), 1 mM EDTA (Sigma) and 16 U/ml Proteinase K (NEB). Cells in lysis buffer were incubated for 1 h at 37 °C, total solution harvested from the plate, and the Proteinase K inactivated by incubating at 95 °C for 10 min. This gDNA solution was stored at −20 °C until use.

**RNA circularization assays.** For all (+)CMV constructs, prior to circularization the 5′ cap was removed as this would otherwise inhibit the circularization reaction. To do so, 12.5 U of RNA 5′ Pyrophosphohydrolase (RppH) was added to 2.5 μl of RNA in 1× ThermoPol® Reaction Buffer (both from NEB) in a 25 μl total reaction volume. These were incubated for 1 h at 37 °C. RNA was then purified from the reactions using a ChargeSwitch® Total RNA Cell Kit (Thermo Fisher Scientific) without lysis incubation or DNase treatment step and with a final elution volume of 12.5 μl. Both decapped (+)CMV and untreated (−)CMV samples were then circularized with reaction conditions as follows. 2 μl of 10x T4 RNA ligase buffer (NEB), 0.1 μl 10 mM ATP (NEB), 1 μl SUPERase In RNase Inhibitor (20 U/μl, Thermo Fisher Scientific), 4 μl 50% PEG8000 (NEB), 1 μl T4 RNA Ligase 1 (10 U/μl, NEB), 10 μl RNA and 1.9 μl $H_2O$.

These reactions were incubated for 4 h at room temperature, then the RNA was repurified using ChargeSwitch® Total RNA Cell Kit as above with 12.5 μl elution volumes for the (+)CMV samples and 25 μl elution volumes for the (−)CMV. Six microliters of the resulting RNA was then reverse transcribed using the QuantiTect® Reverse Transcription (RT) kit (Qiagen) as per the manufacturer's instructions with the specific primer "cRT-CTS2_nest_R" (Supplemental Table 4) and with a 30 rather than a 15-min incubation at 42 °C. A nested PCR was then performed using Phusion® High-Fidelity PCR Master Mix with GC Buffer (NEB) with 500 nM primer concentrations ("cRT-sgRNA_nest_F" + "cRT-CTS2_nest_R" for first PCR, "cRT-sgRNA1_v2_F" + "cRT-CTS2_R" for second PCR, Supplemental Table 4) starting from 1 μl of the cDNA. One microliter of a 1:10 dilution of the first PCR was transferred into the second PCR (20 μl total reaction volume). Cycling conditions were 98 °C for 60 s, 10 (1st PCR) or 20–25 (2nd PCR) cycles of 98 °C for 10 s, 58 °C for 30 s, 72 °C for 15 s, then 5 min at 72 °C.

Following cycling, second PCR products were run on a 2% agarose gels and nucleic acids visualized with GelRed® (Biotium). Gel images were taken using a BioRad GelDoc™ XR+ imager, with exposure times just below what would give saturated pixels (generally between 0.5 and 0.75 s). Image processing was done using GelAnalyzer2010a. First, automatic lane detection was performed, with lanes being manually adjusted in the case of errors. Next automatic peak identification was performed, and again manually curated. Then, rolling ball background subtraction (radius = 25 pixels) was performed. Finally, the raw volume of the correctly processed peak was divided by the sum of the raw volume of all peaks to estimate the processing efficiency.

**T7 endonuclease assays.** The region surrounding the PDL1 cut site from our guide was first amplified from the extracted gDNA using Phusion® High-Fidelity PCR Master Mix with GC Buffer (NEB) with 500 nM primers (forward: TGCTTTTGAATCCTGCACAA, reverse: CCATTGCTAGCCCTTAATCC) in a 30 μl total reaction with 1 μl gDNA. Cycling conditions were 98 °C for 30s, 40 cycles of 98 °C for 10 s, 61 °C for 12 s, 72 °C for 12 s, then 5 min at 72 °C.

Following amplification, products were purified using Agencourt AMPure XP beads as per the manufacturer's instructions with 1.8× sample volumes of beads and a 25 μl final elution volume. 150 ng of product was then resuspended in 19 μl 1× NEBuffer 2 (NEB). These were then heated to 95 °C for 5 min, brought to 85 °C at 2 °C/s, then cooled to 25 °C at 0.1 °C/s. One microliter of T7 Endonuclease I (10 000 U/ml, NEB) was then added and samples incubated at 37 °C for 30 min. Samples were then run on a 2% agarose gels and nucleic acids visualized with GelRed® (Biotium). Gel images were taken using a BioRad GelDoc™ XR+ imager, with exposure times just below what would give saturated pixels (~0.75 s). Image processing was performed using GelAnalyzer2010a. First, automatic lane detection was performed. Next automatic peak identification was performed, and manually curated to include the unedited, and the two edited peaks. Then, peak-to-peak background subtraction was performed. Finally, editing efficiency was calculated as:

$$100 \times \left( 1 - \sqrt{1 - \frac{\text{edited volume}}{\text{total volume}}} \right).$$

**Flow cytometry.** Flow cytometry was performed using either a BD LSRFortessa™ cell analyzer or BD LSRII flow cytometer. ECFP was measured following 405 nm

excitation with a 450/50 bandpass filter. EGFP was measured using a 488 nm excitation with a 530/30 (Fortessa) or 525/50 (LSRII) bandpass filter. iBlue was measured using a 640 nm excitation with a 670/14 bandpass filter. mCherry and mKate (for the intron release system) were measured with a 561 (Fortessa) of 532 nm (LSRII) excitation with a 610/20 bandpass filter.

**Deep-sequencing library preparation.** The total pDNA fraction from the AllPrep DNA/RNA/miRNA Universal Kit (Qiagen) was first treated with 20 U Exonuclease V in NEBuffer 4 supplemented with 1 mM ATP at 37 °C for 1 h to remove genomic DNA. pDNA was then purified using Qiagen MinElute PCR purification columns with a 15 µl elution volume. Triplicate PCR amplifications were then performed for each sample as follows. The reaction mix was composed of 25 µl of 2x GC Phusion Master Mix, 2.5 µl of 10 µM Primer Mix ('cRT-sgRNA1_LIB_F'+'cRT-CTS2_LIB_R', Supplementary Table 4), 18.5 µl H2O, 4 µl purified pDNA. Cycling conditions were 98 °C for 60 s, 25 cycles of 98 °C for 10 s, 58 °C for 30 s, 72 °C for 15 s, then 5 min at 72 °C.

Triplicate amplicons were gel purified using a Qiagen gel extraction kit as per the manufacturer's instructions but with Qiagen MinElute PCR purification columns substituted for Qiaquick columns and a final elution volume of 15 µl buffer EB. Triplicate amplicons were then pooled at equimolar ratios for downstream steps.

**circRNA library preparation.** First, the 5′ cap was removed from the (+)CMV libraries using RppH. For these reactions 37.5 U RppH was added to 750–1000 ng total RNA in 1× ThermoPol buffer (both from NEB) in a 75 µl reaction. These were incubated for 1 h at 37 °C. RNA was then purified from the reactions using ChargeSwitch® Total RNA Cell Kit (Thermo Fisher Scientific) without the lysis incubation or DNase treatment step and with a final elution volume of 25 µl. Both the decapped (+)CMV libraries and the untreated (−)CMV libraries were then circularized as follows. The reaction mixture was composed of 4 µl of 10x T4 RNA Ligase buffer (NEB), 2 µl of 1 mM ATP, 2 µl SUPERase In RNase Inhibitor (20 U/µl, Thermo Fisher Scientific), 8 µl 50% PEG8000 (NEB), 2 µl T4 RNA Ligase 1 (10 U/µl, NEB), 20 µl RNA and 2 µl H2O.

These reactions were incubated at room temperature for 4 h at room temperature. RNA was then repurified using ChargeSwitch® Total RNA Cell Kit (Thermo Fisher Scientific) with a final elution volume of 25 µl. One microgram of circularized RNA was then subjected to reverse transcription using the QuantiTect® Reverse Transcription kit (Qiagen) as per the manufacturer's instructions with the specific primer "cRT-CTS2_nest_R" (Supplemental Table 4) and with a 30 rather than a 15-min incubation at 42 °C. A first PCR amplification was then performed as follows. The reaction mixture was composed of 25 µl 2x GC Phusion Master Mix, 2.5 µl of 10 µM Primer Mix ('cRT_sgRNA_nest_F' + 'cRT_CTS2_nest_R', Supplementary Table 4), 100 ng cDNA, H2O up to 50 µl. Cycling conditions were 98 °C for 60 s, 10 cycles of 98 °C for 10 s, 58 °C for 30 s, 72 °C for 15 s, then 5 min at 72 °C.

PCR products were diluted 1:10 in Ambion® DEPC-treated water (Thermo Fisher Scientific) and used as template for a second PCR as follows. PCR products were diluted 1:10 in Ambion® DEPC-treated water (Thermo Fisher Scientific) and used as template for a second PCR as follows. The reaction mixture was composed of 25 µl 2x GC Phusion Master Mix, 2.5 µl of 10 µM Primer Mix ('cRT-sgRNA1_LIB_F'+'cRT-CTS2_LIB_R', Supplementary Table 4), 1 µl of 1:10 first PCR product, 21.5 µl H2O. Cycling conditions were 98 °C for 60 s, 15 cycles of 98 °C for 10 s, 58 °C for 30 s, 72 °C for 15 s, then 5 min at 72 °C.

Following second PCR all samples were purified using Agencourt AMPure XP beads as per the manufacturer's instructions with 1.8× sample volumes of beads and a 25 µl final elution volume. Triplicate amplicons were then pooled at equimolar ratios for downstream processing.

**Deep-sequencing library indexing and sequencing.** Illumina indices were then added to both purified circRNA and pDNA libraries with another round of PCR amplification. For all samples the D508 was used as a forward index primer, while each sample was given a unique reverse index from D701−D712 (Supplemental Table 5). Reaction conditions were as follows. The reaction mixture was composed of 12.5 µl 2x KAPA HiFi HotStart ReadyMix (Roche) 0.75 µl of 10 µM D508, 0.75 µl of 10 µM Reverse Index Primer, 1 ng of the amplified library and H2O up to 25 µl. The cycling conditions were 98 °C for 3 min, 15 cycles of 98 °C for 20 s, 62 °C for 15 s, 72 °C for 15 s, then 5 min at 72 °C.

Following amplification, products were once again purified using Agencourt AMPure XP beads as per the manufacturer's instructions with 1.8× sample volumes of beads and a 25 µl final elution volume. Libraries were then quantified using a Qubit dsDNA HS (High Sensitivity) Assay Kit (Thermo Fisher Scientific) as per the manufacturer's instructions, and size distributions estimated by 2% agarose gel analysis using GelAnalyzer2010a. All libraries were then pooled and requantified for both concentration and size distribution and diluted to 4 nM. Finally, diluted libraries were sequenced using an MiSeq benchtop sequencer with a 300 cycle Reagent Kit v2 (Illumina) as per the manufacturer's protocols with 10% PhiX DNA spiked in.

**Quantitative RT-PCR.** First, 3 µl RNA from each sample was treated with a TURBO DNA-free™ Kit (Thermo Fisher Scientific) in a 6 µl total reaction as per the manufacturer's instructions to remove residual plasmid contamination. Next, RT was performed using the QuantiTect® RT kit (Qiagen) with a 10 µl reaction volume using the supplied random primer with an incubation time of 30 rather than 15 min at 42 °C. Quantitative PCR (qPCR) reactions were performed on the resulting cDNA using SsoAdvanced™ Universal SYBR® Green Supermix (Biorad) on a CFX384 Touch™ Real-Time PCR Detection System (Biorad). All samples were measured with two primer pairs: "CTS2_qPCR-F" with "sgRNA_common_qPCR-R" for measuring sgRNA abundance, and "dCas9_qPCR-F" with "dCas9_qPCR-R" for dCas9-VP64 abundance as an internal normalization control (primer sequences available in Supplemental Table 4). Primers concentration was 250 nM and annealing temperature was 60 °C.

**Data analysis.** All final analyses were performed in R (version 3.4.1). Statistical testing was performed using the "bayes.t.test" function from the R library "BayesianFirstAid" (version 0.1) with 30,000 iterations for MCMC sampling. BEST tests[17] were used as these are more information-rich than classic *t* tests and are robust to sample distribution and outliers. Tests were paired when relevant and unpaired in other cases (as specified for each test). For flow cytometric analysis a combination of functions from the R packages "flowCore" (version 1.44.2) and custom scripts was used for basic processing (gating, plotting, summary statistics).

Analysis of deep-sequencing data consisted of four stages: (1) pre-processing, (2) variant−barcode association, (3) barcode association refining and trimming, and (4) final analysis. First, the raw FASTQ files were quality trimmed using sickle (version 1.200) in paired-end mode with a quality threshold of 20 and a minimum read length of 70 bp. The (quality trimmed) paired-end reads were then merged into a single read based on sequence overlap using "bbmerge-auto.sh" from BBMap (version 37.48) with $k = 60$ on "strict" mode.

As any read in the circRNA from a properly processed gRNA will not have the tRNA included in the read, an association map between the barcode sequences attached to the gRNAs and the sequence of the associated tRNA was necessary. To create the first iteration of such an association dictionary, we analyzed the plasmid DNA sequencing results using a combination of "ShortRead" (version 1.26.0), "Biostrings" (version 2.36.4), and "plyr" (version 1.8.4) together with custom scripts in R (version 3.2.1). To do so, we first identified reads in the expected amplicon size (between 185 and 200 bp). We next located the barcode sequences using "vcountPattern" and "vmatchPattern" allowing one mismatch (search patterns were "CNNCANNGTNNAGNNNACNN" for the 5′ barcode and "NNGANNNTCNNTCNNGANNGTAA" for the 3′ barcode). Barcode sequences were then retained if the potential barcode hit was <3 bp away from the expected location in the amplicon. Similarly, we identified tRNA sequences with the pattern "GGCTCGTNNGTCTNNNNNNNTGATTCTCGCTTAGGGTGCGAGAGGTCC CGGGNNNNNNNCCCGGACGAGCCC" now allowing three mismatches.

Reads that had identifiable barcodes (5′ and 3′) and a tRNA sequence of correct length and which did not have any ambiguous bases were retained. Next, reads with identical barcode and tRNA sequence were collapsed to obtain read counts. Finally, for any barcode pair with inconsistent tRNA sequences, inconsistent sequences with 2 bp or less difference from the most abundant read and less than 50% of the most abundant read were assumed to be sequencing errors and merged into the most abundant reads count. If more differences were present, the other differing sequences were ignored if they had a read abundance of 10% or less of the most abundant read. If inconsistent reads had more than 2 bp difference from the most abundant read and were >10% the abundance of the most abundant read, all reads of this barcode pair were removed from the dictionary.

Frequencies for each mutation in the overall variant library in each experiment were calculated as the number of reads for that tRNA variant in the retained reads divided by the total number of retained reads. Expected frequency (*f*) for 0 to 10 mutations (*n*) was calculated as $_{16}C_n(f^{(16 − n)}(1 − f)^n$. The mean squared error (m.s.e.) was calculated between the observed frequencies of each number of mutations and the predicted increments of 0.033% across the probable range of values and the best fit chosen which minimized m.s.e. The distribution of mutations across nucleotides was calculated as the number of reads with a mutation at a given site divided by the total number of reads.

For the circRNA, all merged reads with a total length of 50 bp or more were considered. We next identified the tRNA, 5′ and 3′ barcodes as was done for the pDNA for the (−)CMV libraries. For (+)CMV libraries we required only the first nine variable bases of the 5′ barcode to be present (a pattern of "CGGTGCNNCANNGTNNAGNNN" for the 5′ barcode) as the barcode was partially truncated in a majority of the sequences, possibly due to degradation during the 5′ cap removal. In the case of the truncated requirement, pDNA barcode associations were also collapsed to retain only those with unique truncated barcode to tRNA sequence associations. Following identification of barcodes and tRNA sequences, we next classified each read based on its degree of processing. To be classified as processed, a read had to have a total length less than 150 bp, have one but not the other barcode sequence (i.e. either 5′ or 3′), and not have an identifiable tRNA sequence. Reads were classified as fully unprocessed if both barcodes and tRNA sequence were present and the overall length was 150 bp or greater. Finally,

they were classified as partially processed if total length was 150 bp or greater, one barcode was present and a tRNA sequence.

We next created barcode-tRNA sequence dictionaries based on the unprocessed circRNA as was done for the pDNA. In this case dictionaries were separate for the 5′ and 3′ barcodes as in the case of partial processing both barcodes were not necessarily present in the same read. To obtain our final barcode-tRNA dictionary, we then compared the circRNA dictionaries to the pDNA dictionaries. In cases of disagreement these were resolved as previously done within the pDNA or circRNA (i.e. merging those which were very similar, ignoring very low abundance disagreeing reads, and removing barcodes with ambiguous associations). Finally, we only retained those associations which had been observed in at least three reads and had at least one processed and one unprocessed read in the circRNA dataset.

Processing efficiency was calculated for each tRNA variant as $100 \times$ (processed reads/all reads). To determine the processing ability of each mutation a binomial distribution was inferred for each replicate based on the number of processed reads and the number of total reads using the R function "dbinom" in the range of 1–100% with 1% increments. These distributions were averaged across the three replicates to get an overall probability distribution. The maximum likelihood of the combined binomial distributions was used as the estimated processing efficiency. Significance values were calculated by determining the area of overlap of the two probability distributions to be compared. The effect of each mutation on promoter activity was calculated as $\log_2(cRNA_{freq}/pDNA_{freq})$. Statistical testing for differences in promoter activity were calculated using paired BEST tests comparing each mutation to wild type.

**Reporting summary**. Further information on experimental design is available in the Nature Research Reporting Summary linked to this article.

## Code availability
Custom R scripts used to analyze HTS data are available from the corresponding authors upon reasonable request.

## Data availability
Raw HTS data (Fastq files) have been deposited into the Sequence Read Archive (SRA). SRA accession: PRJNA521493. All other raw data that are not found in the supplementary information are available from the corresponding authors upon reasonable request. Relevant plasmids described in this study are available from Addgene (http://www.addgene.org/Tudor_Fulga/).

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

## Acknowledgements
D.J.H.F.K. is funded by a CIHR Postdoctoral Fellowship. T.A.F. was supported by MRC (G0902418), BBSRC (BB/N006550/1), and Wellcome Trust ISSF (105605/Z/14/Z). Y.S. M. is funded by the WIMM prize studentship, Clarendon Scholarship and Christopher Welch Scholarship. Q.R.V.F. was funded by a Wellcome Trust Ph.D. Studentship. M.J. acknowledges funding from the University of Oxford and the EPSRC & BBSRC Centre for Doctoral Training in Synthetic Biology (grant EP/L016494/1). T.A.M. is funded by Medical Research Council (MRC, UK) Molecular Haematology Unit Grant MC_UU_12009/6.

## Author contributions
D.J.H.F.K. and T.A.F. conceived the project, designed the experiments, interpreted the results and wrote the manuscript. D.J.H.F.K. performed all experiments. Y.S.M. helped with the tRNA screen design and analysis framework. M.J. assisted with cloning and qPCR optimization. Q.R.V.F. assisted with the design of the reporter assays and plasmid preparation. H.B. cloned wild-type tRNA^Gln constructs. T.A.M. provided project guidance. All authors provided input on the manuscript.

## Additional information

**Competing interests:** T.A.M. is one of the founding shareholders of Oxstem Oncology (OSO), a subsidiary company of OxStem Ltd. All other authors declare no competing interests.

