## [Peer Review File · Nature Communications]

Reviewers' Comments:

Reviewer #1:

Remarks to the Author:

Due to its endogenous 5' and 3' processing, tRNA has been used to generate multiple gRNAs from a single transcript which may be expressed under the control of an inducible Pol-II promoter. However, tRNA contains intrinsic Pol-III promoter activity which leads to the constitutive expression of gRNAs. In order to control spatial/temporal expression and improve ON-OFF ratios of gRNA induction, this study aims to dissociate the processing and promoter activities of human tRNAs. The authors have developed a mutant screening strategy and identified some important mutations that affect tRNA promoter activity or its 5'/3' processing efficiencies. They have successfully created new tRNA variants that have no detectable promoter activity but retain sufficient 5'/3' processing efficiencies. The new tRNA variant system was better than most alternatives, and was equivalent to the optimized Csy4-based system for Pol-II promoter driven production of functional gRNAs. The experiments involved in this study are well designed and appears to be carefully performed. The resulting new tRNA variant is a significant improvement to the existing tools for inducible gRNA production and spatial/temporal control of CRISPR/Cas-enabled genome editing.

Reviewer #2:

Remarks to the Author:

This study by Knapp, Fulga and colleagues describes the identification and evaluation of a rationally designed tRNA backbone for CRISPR genome engineering applications. Inclusion of flanking tRNAs has been shown to be an effective means to facilitate sgRNA multiplexing and expression from conditional RNA Pol II promoters. However, tRNAs are known to process RNA Pol III promoter activity, which can potentially interfere with upstream Pol II promoters.

This study therefore aimed to identify tRNA designs that can be efficiently excised from a longer transcript, but do not function as effective Pol III promoters. To this end the authors devised a screening system that allows measuring the abundance of various processed fragments of a tRNA:sgRNA array. They combine this assay with a library of tRNA mutants to measure the effect of point mutations at various positions on promoter activity, as well as the efficiency of 5' and 3' excision mediated by RNase P and Z. This revealed some mutations that appeared to reduce promoter activity but left processing relatively unaffected. However, when these mutations were retested in pairwise combinations, they were often found to have the opposite effect (i.e. lead to reduced processing, but retain significant promoter activity). Why these mutations have opposing effects in the different experiments remains unclear. In the aforementioned experiments the authors also include some tRNA mutants bearing larger deletions, which match the set criteria much better. The authors therefore focus on these variants and compare them to a number of other strategies to produce functional sgRNAs from Pol II promoters.

Major point:

While this study is certainly interesting, I am not convinced that it makes a compelling case that the presented tRNA designs will be generally useful for typical genome engineering applications. This study converges on the delC55G design, which leads to a dramatic reduction in activity when compared to the conventional U6-sgRNA expression system (retaining activity in only ~1% of cells in Fig. 3e and 4b). Moreover, 'activity' is measured in a highly artificial system containing 8 binding sites for the same validated sgRNA. In contrast, most real life genome editing applications rely on the activity of sgRNAs with a single binding site at the target locus. Whether the levels of functional sgRNAs coming from a delC55G construct would be sufficient under such circumstances remains unknown. In order to demonstrate the general usefulness of the delC55G design, the authors would need to show the suitability of delC55G flanked sgRNAs for CRISPRn and CRISPRa

applications targeting endogenous genes. To evaluate how generalizable such an approach would be, this should also involve constructs harboring different Pol II promoters, as promoter strength is expected to have a large influence on gene editing efficiency.

Other points:

The authors do not explain in detail the need for conditional sgRNA expression. Given that CRISPR is a facultative two component system, one would imagine that simply placing Cas9 under control of a specific Pol II promoter would suffice for spatial or temporal control of genome editing. Why this might not be the case or what additional benefits Pol II driven sgRNA expression might have, should be explained more explicitly.

Furthermore, I am aware of at least one study (current reference 13), which experimentally demonstrates that tissue-specific expression of Cas9 is not sufficient to achieve tissue-specific genome editing. However, in this context full length wildtype tRNAs were sufficient to facilitate conditional genome editing, suggesting that rationally designed tRNA variants without promoter activity are not always necessary. It is therefore essential to explain better the overall rationale of this study. Of course, it would be highly desirable if the need for tRNA variants would be experimentally demonstrated.

Throughout the manuscript the authors plot the activity levels of the various constructs on a log₁₀ scale. While good at visualizing subtle differences in activity of the different designs, these graphs are much less good at showing the much stronger differences in activity compared to the unmodified tRNAs or U6 constructs. I would argue that these differences are of greater practical importance, whereas it is less interesting to see small differences between constructs that can all be described as poorly active. Use of a linear scale would reflect the data better.

Reviewer #1 (Remarks to the Author):

[...] The experiments involved in this study are well designed and appears to be carefully performed. The resulting new tRNA variant is a significant improvement to the existing tools for inducible gRNA production and spatial/temporal control of CRISPR/Cas-enabled genome editing.

We thank the reviewer for their endorsement of our technology and manuscript. To further demonstrate the utility of this system, we have now added additional data on editing genomic loci and activation of endogenous genes. We have also streamlined the cloning process and will deposit all relevant constructs with *Addgene* to improve ease-of-use for other groups. We hope that our system will be a useful tool to the genome engineering scientific community.

Reviewer #2 (Remarks to the Author):

Major point:

1. While this study is certainly interesting, I am not convinced that it makes a compelling case that the presented tRNA designs will be generally useful for typical genome engineering applications. This study converges on the delC55G design, which leads to a dramatic reduction in activity when compared to the conventional U6-sgRNA expression system (retaining activity in only ~1% of cells in Fig. 3e and 4b). Moreover, 'activity' is measured in a highly artificial system containing 8 binding sites for the same validated sgRNA.

We agree with the reviewer that the degree of activation is an important parameter when using inducible systems. While U6 or other Pol-III promoters indeed have higher activity, they are not inducible, and thus not suitable for applications which require multiple editing events to be controlled independently. Unlike the expression of proteins under inducible Pol-II promoters, all currently existing systems for Pol-II-mediated gRNA production display substantially weaker activity than U6, as demonstrated by our comparative analysis. However, this does not preclude their utility for genome engineering applications. We would also like to emphasize that in our experiments we were measuring a snapshot of activity, and thus over a longer timeframe cumulative effects such as editing could in fact reach higher percentages (see point 2 below).

Regarding our detection method, the reporter system containing multiple binding sites is a well-established framework to study the activity of CRISPR gRNAs at single cell resolution with high sensitivity (see Farzadfard et al., *ACS Synth Biol.* 2013; Nissim et al. *Mol. Cell* 2014; Shechner et al., *Nature Methods* 2015; Ferry et al. *Nat. Comm.* 2017). Since this reporter is not affected by biological (e.g. epigenetic state, microRNA) and technical (eg. non-specific antibody binding) confounding factors, we reasoned that it was the optimal system for assessing the impact of different tRNA mutations on the production of functional gRNAs. We have expanded in the revised manuscript the rationale for using this strategy to assess the level of functional gRNAs expressed from our modified tRNA scaffolds. In addition, we now show that our tRNA system is compatible with both transcription activation and editing of endogenous genes (see below).

2. [...]. In order to demonstrate the general usefulness of the delC55G design, the authors would need to show the suitability of delC55G flanked sgRNAs for CRISPRn and CRISPRa applications targeting endogenous genes.

We completely agree with the reviewer on the importance of demonstrating the implementation of this technology using endogenous targets. Following the reviewer's suggestion, we have now performed both endogenous activation (CRISPRa) and editing

(CRISPRn) experiments using combinations of the best performing tRNA scaffolds targeting functionally relevant genes (NGFR and PDL1).

With regard to CRISPRa activity, in the presence of Pol-II induction we observed a significant increase in NGFR levels for all tRNA scaffold combinations with the exception of wild-type tRNA^{Pro} (see Fig. 4a, b in the revised manuscript). However, the ON/OFF ratios are lower than what we observed in the reporter system. This is likely due to the presence of background NGFR staining, thus reducing our sensitivity of detection. Importantly, in the editing assays we obtain up to 10% efficiency after 4 days, as would be expected from a cumulative effect of 1-3 % active gRNAs/day observed in the reporter assays (see Fig. 4c, d and Supplementary Figure 7 in the revised manuscript). Furthermore, these numbers are likely an underestimate as they include at least 20% untransfected cells (based on similar conditions used for CRISPRa experiments).

These results suggest that for uses such as lineage specific guide activation, the tRNA system will allow substantial amounts of cutting over time and thus provide a very useful method, particularly where multiple lineages require differential editing. To facilitate the use of these novel tRNA scaffolds, we have created a mathematical model for predicting cumulative editing events over time for various gRNA production efficiencies (see Supplementary Figure 8 in the revised manuscript).

3. To evaluate how generalizable such an approach would be, this should also involve constructs harboring different Pol II promoters, as promoter strength is expected to have a large influence on gene editing efficiency.

The reviewer correctly points out that promoter activity will likely play an important role in determining gRNA levels and thus editing efficiency. Following their suggestion, we have now added data to the editing assays using PGK, a weaker promoter compared to CMV (Qin et al. PLoS One 2010) (see Fig. 4c, d and Supplementary Figure 7 in the revised manuscript). As expected, in the presence of PGK the efficiency was markedly reduced. However, the observed ~1% editing efficiency fits with expectations (see model in Supplementary Figure 8 of the revised manuscript) given that in HEK293T cells PGK-driven expression is ~ ¼ that of CMV (Qin et al. PLoS One 2010). This data indicates that selection of promoters with strong activity will likely be an important consideration when applying this system. Nonetheless, weak promoters could be used, although in these cases longer time periods will likely be required to achieve meaningful levels of editing. These points have now been discussed in the revised manuscript.

Other points:

1. The authors do not explain in detail the need for conditional sgRNA expression. Given that CRISPR is a facultative two component system, one would imagine that simply placing Cas9 under control of a specific Pol II promoter would suffice for spatial or temporal control of genome editing. Why this might not be the case or what additional benefits Pol II driven sgRNA expression might have, should be explained more explicitly.

We apologise for not explaining more clearly the rationale for developing our system for conditional gRNA expression. Placing the Cas9 under a specific Pol-II promoter could enable spatial-temporal control of genome editing. However, there are several caveats to this strategy: *i)* this approach only allows single lineage/time/set of gRNAs to be controlled; *ii)* given the extremely high efficiency of Cas9 nucleases, even very low background promoter activity can result in significant unintended editing events (see Port and Bullock *Nat Methods*, 2016). In contrast, by placing gRNAs under Pol-II control, an arbitrary number of lineages can be differentially edited thus greatly expanding the spatial/temporal control of Cas9 activity. We have now revised the manuscript to clarify this point.

2. Furthermore, I am aware of at least one study (current reference 13), which experimentally demonstrates that tissue-specific expression of Cas9 is not sufficient to achieve tissue-specific genome editing. However, in this context full length wildtype tRNAs were sufficient to facilitate conditional genome editing, suggesting that rationally designed tRNA variants without promoter activity are not always necessary. It is therefore essential to explain better the overall rationale of this study. Of course, it would be highly desirable if the need for tRNA variants would be experimentally demonstrated.

Although we agree with the reviewer that in the study by Port and Bullock wildtype tRNAs were sufficient to achieve tissue-specific editing, it should be noted that this activity required Cas9 to be under the control of the same inducible promoter. Critically, this study was also performed in *Drosophila*, and as the authors stated in Supplementary Figure 2 legend, it is unlikely that their system would function in mammalian cells (“*Autonomous RNA pol III promoter activity of tRNAs can drive sgRNA expression in mammalian cells. The absence of appreciable UAS-t:sgRNA-wls2x activity without Gal4 (see also Fig. 2d, e) indicates that if such a mechanism exists in Drosophila, it is not sufficient to induce the levels of sgRNA needed for mutagenesis.*”).

This limitation provided in fact the rationale for engineering tRNA scaffolds that lack promoter activity but retain processing capabilities. Therefore, our study was necessary for the translation of tRNA-based gRNA release systems into biomedically relevant mammalian cells. Furthermore, even in *Drosophila*, the requirement of having Cas9 on the same promoter precludes independent control of multiple edits/lineages, which in principle could be achieved using these novel tRNA scaffolds. A more thorough explanation of these points has now been added to the Introduction.

3. Throughout the manuscript the authors plot the activity levels of the various constructs on a log₁₀ scale. While good at visualizing subtle differences in activity of the different designs, these graphs are much less good at showing the much stronger differences in activity compared to the unmodified tRNAs or U6 constructs. I would argue that these differences are of greater practical importance, whereas it is less interesting to see small differences between constructs that can all be described as poorly active. Use of a linear scale would reflect the data better.

Although we understand the reviewer’s concern, we point out that the data are *log*-normal and span several orders of magnitude. A *log* display is more informative of the total distribution of the data and thus a more appropriate approach from a statistical perspective. Furthermore, based on both our data and all currently published systems, Pol-III promoters are much stronger than Pol-II. Considering that our primary interest was to develop a Pol-II-based system, we argue that a linear display would unduly compress the data of interest while giving too much emphasis to a control condition used only to reflect the upper limit of activity. As semi-log markers are provided, linear values can still be easily inferred from the existing plots, and thus we have opted to retain the more informative *log*-plotting.

Reviewers' Comments:

Reviewer #2:

Remarks to the Author:

I would like to thank the authors for their thoughtful answers to my points. I am particularly pleased that the authors have added essential data to the manuscript that tests the described tRNA variants on endogenous target genes. These experiments establish that the tools described in this study in principle retain enough activity to activate and mutagenise endogenous genes, although the activity is low, which is in line with the data obtained using the reporter system. While activity is likely to be a major consideration for many applications and hence adoption of this method might be limited, I concur that there can be important applications where spatial or temporal specificity is the major concern. As such, this manuscript makes a significant contribution to the field.